# Navigating alcohol's impact: A mixed-methods analysis of community perceptions and consequences in Northern Tanzania

**Alena Pauley**[1], **Madeline Metcalf**[1], **Mia Buono**[1], **Kirstin West**[1], **Sharla Rent**[1,2], **William Nkenguye**[3,4], **Yvonne Sawe**[3], **Mariana Mikindo**[3], **Joseph Kilasara**[3,5], **Bariki Mchome**[3,6], **Blandina T. Mmbaga**[1,3,7], **João Ricardo Nickenig Vissoci**[1,8], **Catherine A. Staton**[1,8] *

**1** Duke Global Health Institute, Duke University, Durham, North Carolina , United States of America, **2** Duke Department of Pediatrics, Duke University, Durham, North Carolina, United States of America, **3** Kilimanjaro Christian Medical Centre, Moshi, Tanzania, **4** Kilimanjaro Christian Medical University College, Moshi, Tanzania, **5** Department of Clinical Nursing, Kilimanjaro Christian Medical University College, Moshi, Tanzania, **6** Department of Obstetrics and Gynecology, Kilimanjaro Christian Medical University College, Moshi, Tanzania, **7** Kilimanjaro Clinical Research Institute, Moshi, Tanzania, **8** Duke Department of Surgery, Duke University Medical Center, Durham, North Carolina, United States of America

* catherine.lynch@duke.edu

## Abstract

### Background

Worldwide, alcohol is a leading risk factor for death and disability. Tanzania has particularly high rates of consumption and few resources dedicated to minimizing alcohol-related harm. Ongoing policy efforts are hampered by dynamic sociocultural, economic, and regulatory factors contributing to alcohol consumption. Through the voices of Kilimanjaro Christian Medical Centre (KCMC) patients, this study aimed to investigate community perceptions surrounding alcohol and the impact of its use in this region.

### Methods

This mixed-methods study was conducted at KCMC between October 2021 and May 2022. 676 adult (≥18 years old) Kiswahili-speaking patients who presented to KCMC's Emergency Department (ED) or Reproductive Health Clinic (RHC) were enrolled through systematic random sampling to participate in quantitative surveys. Nineteen participants were selected for in-depth interviews (IDIs) through purposeful sampling. The impact and perceptions of alcohol use were measured through Drinkers' Inventory of Consequences (DrInC) scores and analyzed in RStudio using means and standard deviations. IDI responses were explored through a grounded theory approach using both inductive and deductive coding methodologies in NVivo.

**Data availability statement:** Data are only available upon reasonable request, as participants did not consent to public data publishing, and data transfer requires a written agreement approved by Kilimanjaro Christian Medical Centre Ethics Committee and the National Institute for Medical Research (Tanzania). Data are available upon request from Kilimanjaro Christian Medical Centre Ethics Committee and the National Institute for Medical Research (Tanzania) representative Gwamaka W. Nselela via email (gwamakawilliam14@gmail.com) for researchers who meet the criteria for access to confidential data.

**Funding:** This project was funded by the Duke Global Health Institute Graduate Student funds (AMP), and the Josiah Trent Foundation (21-06 to CAS). These two financial awards funded the salaries of JK, YS, and MMi as research assistants hired specifically for this study. WN's time and participation was made possible by "The TRECK Program: Trauma Research Capacity Building in Kilimanjaro, Tanzania" (D43TW012205 to CAS". No other authors received specific funding for this work. Infrastructure built by an NIH grant (R01 AA027512 to CAS) was used to support the data collection process for this grant to understand gender-related aspects of alcohol use at KCMC. The funders had no role in study design, data collection and analysis, decision to publish, or preparation of the manuscript.

**Competing interests:** The authors have declared that no competing interests exist.

## Results

Men attending the ED were found to have the highest average [SD] DrInC scores (16.4 [19.6]), followed by ED women (9.11 [13.1]), and RHC women patients (5.47 [9.33]), with higher scores indicating a broader array of consequences. Participants noted alcohol to have both perceived advantages and clear harms within their community. Increased conflict, long-term health outcomes, financial instability, stigma, and sexual assault were seen as negative consequences. Benefits were primarily identified for men and included upholding cultural practices, economic growth, and social unity. Physical and financial harm from alcohol impacted both genders; however, alcohol-related stigma and sexual assault were found to affect women disproportionately.

## Conclusion

Our findings suggest that perceptions around drinking are nuanced, and alcohol's social and physical consequences differ significantly by gender. To effectively minimize local alcohol-related harm, future alcohol-focused research and policy efforts should consider the complex sociocultural role that alcohol holds in the Moshi community.

## Introduction

Alcohol consumption has been implicated as a leading cause of harm, disability, and death worldwide [1]. The harmful use of alcohol is a growing and avoidable cause of premature disability and death, each year causing 5.1% of all disability-adjusted life years (DALYs) and 3 million deaths globally [1]. Alcohol use has been linked to over 200 adverse medical conditions, including digestive diseases, infectious diseases, cardiovascular disease, certain cancers, and injuries [2–4], and has also been associated with poor employment outcomes [5], individual and societal economic harm [4,6–8], social stigma [9,10], and sexual violence [11–13].

While hazardous alcohol consumption is a global issue, its use is increasing especially rapidly in low- and middle-income countries [14–16]. For example, alcohol use has become the leading risk factor for disease burden in many sub-Saharan African countries, including Tanzania [17,18]. As low-income populations typically have fewer resources, financial savings, and poorer access to health care services, alcohol-related harm in these contexts is more devastating both physically and financially [1,19–21]. Although alcohol use is associated with significant harms, it remains deeply embedded in cultural and social practices around the world [22]. In Tanzania, for example, alcohol use is often normalized from a young age and used as a form of social currency—practices that contribute to higher levels of consumption [23].

Adding greater complexity to alcohol's impact on individuals and communities, alcohol consumption has been shown to differ in populations according to cultural norms, regulations, age, and socio-economic status [24]. To address the factors

influencing alcohol consumption, the World Health Organization (WHO) created the Conceptual Model of Alcohol Consumption and Health Outcomes. The model aims to decrease the morbidity and mortality that follow harmful alcohol use patterns and demonstrates that understanding an individual country's sociocultural context is imperative to alter alcohol use behaviors at an individual and country level [25].

Moshi, located near Mount Kilimanjaro National Park in Tanzania, has one of the highest reported rates of alcohol use in the country, with a prevalence 2.5 times higher than in nearby regions [26,27]. Previous studies have shown that 60% of Moshi residents frequently consume alcohol, and 23% suffer from alcohol use disorder [28]. Among injury patients presenting to the Kilimanjaro Christian Medical Centre (KCMC) Emergency Department (ED), nearly 30% had been consuming alcohol when the injury occurred [33]. In addition to the trauma burden associated with alcohol-related injuries in Moshi, alcohol consumption has also been linked to social stigma, ostracization around female alcohol users [29], and unsafe sexual behaviors [30,31]. While alcohol-related harms have been identified in Moshi, current literature exploring how community members view the harms and benefits of alcohol is sparse.

Previous work from our team has examined alcohol-related stigma [29,32], perceptions of alcohol use [23], the prevalence of alcohol-related injuries upon arrival to the ED [33], and the development and implementation of interventions to reduce alcohol misuse for the ED injury patient population [34,35]. However, an exploration of the specific societal views of alcohol use within Moshi or the perceived harms and benefits that underlie continued consumption has yet to be conducted. Understanding the full scope of alcohol's impact and role within the community is essential in combatting the negative effects of intake as it allows local leaders and researchers to address the root cause of dependence. The justification of this work lies in its potential ability to inform future alcohol-reduction policies, interventions, and research initiatives, allowing them to be more targeted, socioculturally appropriate, and effective in decreasing alcohol use and related harm in Moshi [36]. This development process can also serve as a model for population-specific harm reduction interventions in other settings. Through the voices of patients at KCMC this analysis addresses this gap by investigating the societal perceptions of alcohol along with the negative and positive ways that alcohol use impacts individuals and the broader Moshi community.

## Methods

### Study overview and setting

This hospital-based, cross-sectional study, conducted between October 2021 and May 2022, used a sequential, explanatory, mixed-methods approach to explore perceptions and implications of alcohol use among ED and Reproductive Health Center (RHC) patients at KCMC. This mixed-methods approach was chosen to facilitate the triangulation of data so that robust and accurate epidemiological conclusions could be drawn while also providing a deeper cultural context for the complex social, economic, and physical implications of alcohol use in Moshi.

KCMC, which is located near the border of Kenya and Kilimanjaro National Park, was chosen as the research site as it is the largest referral and teaching hospital in the Kilimanjaro region, has existing research infrastructure, and is in an area known to have high rates of alcohol use. Data was collected at two clinic departments: the ED and RHC. The ED was chosen to better facilitate an understanding of risky alcohol use behaviors, given the correlation between excessive alcohol use and injuries or accidents that necessitate emergency care [32,37,38]. Additionally, previous work from our team in developing alcohol-reduction interventions at KCMC's ED meant that continued work in this unit would align with programmatic goals and increase the likelihood of intervention success [39]. The RHC, on the other hand, was selected because of its primarily female patient population, which facilitated a richer, gynocentric perspective on alcohol use, which has been under-represented in past local research. Measuring alcohol use in pregnant or postpartum women seeking gynecological care was also important for maternal and child health, given the high rates of alcohol use during pregnancy previously described in this setting [40,41]. As such, enrolling patients from these two clinical units allowed us to better explore the intersection of gender and hazardous drinking, the primary goal of the overarching study from which this data originates.

## Study population

All enrolled patients met the following eligibility criteria: 1) fluency in Kiswahili, 2) ability to provide informed consent, 3) 18 years of age or older, 4) not a prisoner, and 5) received initial care at KCMC's ED or RHC. Perceived or measured alcohol use was not considered when determining patient eligibility, given that the study intended to provide a representative sample from the ED and RHC. The ability to provide informed consent for participation in the study was defined as being medically stable, clinically sober from alcohol, and able to complete the verbal survey independently. Upon initial approach, patients who did not meet these criteria were reevaluated for participation within 24 hours of arrival at KCMC or before discharge. Prior to administering the survey, all patients verbally confirmed that they had not been previously enrolled in the study.

## Quantitative data

**Study design.** Cross-sectional, quantitative survey data assessing demographic factors, alcohol use, and alcohol-related consequences were collected from KCMC ED and RHC patients. As discussed below, preliminary quantitative analyses were used to guide our qualitative sampling strategy.

**Sample size estimation.** Quantitative sample size was calculated based on the difference in average Alcohol Use Disorder Identification Test (AUDIT) scores between our RHC and ED patient populations, as the prevalence of alcohol use disorder (AUD) was a primary point of interest in the broader study [42]. Based on existing literature and the experience of local research team members, we hypothesized the prevalence of AUD to be 10% among RHC women, 15% among ED women, and 30% among ED men. Thus, at 80% power and with 90% confidence, a sample size of 587 participants was calculated to sufficiently estimate the prevalence of risky drinking between all three study subgroups. Given that the expected difference in the prevalence of AUD was narrower between the two female populations than between either male and female groups, the study team initially prioritized enrolling more women to ensure sufficient statistical power to detect differences between the two female populations.

Preliminary data analysis revealed that the prevalence of AUD was much higher than anticipated (40% in ED women and 45% in ED men). Recalculations of sample size based on these updated prevalence estimates revealed that 1,200 patients would need to be enrolled to determine a difference in proportions between these two groups. While the study timeline and funding could not accommodate the newly estimated sample size of 1200 participants, with IRB approval, the study's target sample size was increased to enroll as many male and female participants as feasible within the study timeline.

**Sampling technique.** Over the course of the study timeline, 676 patients were enrolled through a systematic random sampling strategy, with 655 participants fully completing survey questionnaires. A systematic random sampling strategy was employed to acquire a representative sample of our three patient populations, male ED, female ED patients, and female RHC patients, and better describe the characteristics of each. Except for ED females, every third patient was approached for potential study participation using the ED triage or RHC intake registries. For women seeking ED care, every eligible individual was approached as significantly fewer women than men were ED patients, and more women were needed to accurately determine the prevalence of AUD.

**Data collection.** KCMC ED and RHC patients were approached for participation in this study by a Kiswahili-speaking, Tanzanian research assistant of the same gender (i.e., female research assistants approached female patients) trained in good clinical practices and specific study procedures.

After the research assistants ensured patients were medically stable, they approached patients in a quiet, private location and introduced the study. All patients had the opportunity to decline participation, but if willing to proceed, an in-depth discussion of the risks and benefits of study participation was provided. If agreeing to participate, written consent was obtained, and survey questions were delivered while within the same private, secure location. All questions were administered orally by a research team member to encourage participation from all literacy levels.

**Instruments and variable measurements.** Quantitative surveys consisted of three major components: (1) collection of basic demographic data, such as participants' age and gender; (2) assessment of typical alcohol use practices, including frequency and quantity of consumption as well as typical expenditure on alcohol; and (3) administration of validated instruments, specifically the AUDIT and Drinker Inventory of Consequences (DrInC) tools. The AUDIT is a 10-item screening tool developed by the WHO to identify hazardous, harmful, and dependent patterns of alcohol use [43,44]. Scores range from 0 to 40, with higher scores indicating more hazardous consumption and scores of 8 or greater signaling a clinically significant risk for adverse alcohol-related health effects [45–48]. The AUDIT has undergone extensive psychometric and clinical validation across diverse populations and has been specifically adapted and validated in Tanzania in KiSwahili, ensuring cultural and linguistic appropriateness for this context [49,50].

The DrInC was used to quantitatively assess the consequences of alcohol use among participants. This 50-item, yes/no questionnaire has been previously cross-culturally adapted and clinically validated at KCMC [51]. The tool evaluates alcohol-related consequences across five domains: Physical, Social Responsibility, Interpersonal, Intrapersonal, and Impulse Control (Table 1) [52,53]. Of the 50 items, 45 assess negative consequences and are positively scored, while 5 serve as control items, reflecting perceived benefits of alcohol use (e.g., enjoying the taste or drinking without problems), and are reverse scored. Responses were collected for both the past three months and across the participant's lifetime. Prior confirmatory factor analysis conducted in the study setting demonstrated the construct validity of the tool, confirming its five-domain structure as reflective of distinct alcohol-related consequences [51]. Reliability was also supported by high internal consistency, with Cronbach's alpha exceeding 0.8 across all domains [51]. These findings validate the tool's suitability amongst KiSwahili-speaking participants, ensuring cultural and linguistic relevance in this context.

DrInC scores range from 0 to 50, with higher scores overall indicating more affirmative responses to various alcohol-related items as shown in Table 1 below. Although the DrInC tool quantifies a range of alcohol-related consequences, it does not measure each item's severity or frequency, and no universally accepted clinical cutoff score exists. Designed to capture the broad spectrum of alcohol-related issues, the tool has not been associated with specific cutoff scores in previous studies [51–53]. Consistent with this approach, we did not apply a cutoff score in our study, and instead, DrInC scores were analyzed in relation to alcohol use patterns as described below.

**Data analysis procedure.** AUDIT and DrInC scores for the three patient populations were assessed through means and standard deviations for overall and category-specific scores. The relationship between patient DrInC and AUDIT scores was measured through a linear regression analysis to identify if an association between increased alcohol consumption and more consequences existed, with the independent variable being DrInC scores and the dependent variable being AUDIT scores. Incomplete surveys were still included in the analysis. All quantitative data were analyzed in R Studio version 1.4.

## Qualitative data

*Study design.* Using a purposive sampling strategy guided by preliminary quantitative data analyses, 19 individuals who had previously been enrolled in this study and completed all study questionnaires were contacted either at the end of their survey or later by phone to gauge their interest in participating in a semi-structured IDI. IDIs were selected as the study's qualitative method due to the highly sensitive and stigmatizing nature of the study topic, particularly for women participants. This one-on-one format facilitated open discussions, ensuring participants felt comfortable sharing their experiences while preserving privacy and confidentiality. All contacted patients agreed to participate. IDIs were conducted at KCMC and later analyzed using a grounded theory methodology.

*Sample size estimation.* Of the 676 individuals initially enrolled in this study, all 655 patients who had fully completed quantitative surveys were eligible for IDI enrollment. The study team aimed to enroll 20 IDI participants — ten male participants from the ED group and ten female participants, five each from the ED and RHC groups — or until data saturation was reached, whichever occurred first. This approach was chosen to provide equal representation from both genders, while also prioritizing thematic saturation as the primary determinant for concluding data collection.

**Table 1. Consequence Domains and Corresponding Items of the Drinkers' Inventory of Consequences.**

| DrInC Subcategories | Corresponding Items |
|---|---|
| Physical | "I have had a hangover after drinking" <br> "After drinking, I have had trouble with sleeping, staying asleep, or nightmares" <br> "I have been sick and vomited after drinking" <br> "Because of my drinking, I have not eaten properly" <br> "My physical health has been harmed by my drinking" <br> "My physical appearance has been harmed by my drinking" <br> "My sex life has suffered because of my drinking" <br> "While drinking or intoxicated, I have been physically hurt, injured, or burned" |
| Social Responsibility | "I have missed days of work or school because of my drinking" <br> "The quality of my work has suffered because of my drinking" <br> "I have failed to do what is expected of me because of my drinking" <br> "I have gotten into trouble because of drinking" <br> "I have had money problems because of my drinking" <br> "I have spent too much or lost a lot of money because of my drinking" <br> "I have been suspended/fired from or left a job or school because of my drinking" |
| Interpersonal | "My family or friends have worried or complained about my drinking" <br> "My ability to be a good parent has been harmed by my drinking" <br> "While drinking, I have said or done embarrassing things" <br> "While drinking, I have said harsh or cruel things to someone" <br> "My marriage or love relationship has been harmed by my drinking" <br> "My family has been hurt by my drinking" <br> "A friendship or close relationship has been damaged by my drinking" <br> "My drinking has damaged my social life, popularity, or reputation" <br> "I have lost a marriage or a close love relationship because of my drinking" <br> "I have lost a friend because of my drinking" |
| Intrapersonal | "I have felt bad about myself because of my drinking" <br> "I have been unhappy because of my drinking" <br> "I have felt guilty or ashamed because of my drinking" <br> "When drinking, my personality has changed for the worse" <br> "I have lost interest in activities and hobbies because of my drinking" <br> "My spiritual or moral life has been harmed by my drinking" <br> "Because of my drinking, I have not had the kind of life that I want" <br> "My drinking has gotten in the way of my growth as a person" |
| Impulse Control | "I have driven a motor vehicle after having three or more drinks" <br> "My drinking has caused me to use other drugs more" <br> "I have taken foolish risks when I have been drinking" <br> "When drinking, I have done impulsive things that I regretted later" <br> "I have gotten into a physical fight while drinking" <br> "I have smoked more when I am drinking" <br> "I have been overweight because of my drinking" <br> "I have been arrested for driving under the influence of alcohol" <br> "I have had trouble with the law (other than driving while intoxicated) because of my drinking" <br> "I have had an accident while drinking or intoxicated" <br> "While drinking or intoxicated, I have injured someone else" <br> "I have broken things or damaged property while drinking or intoxicated" |

This original estimated sample size of 20 participants was informed by commonly accepted recommendations in qualitative research, which suggest 15–20 interviews as a general guideline for reaching saturation in relatively homogenous groups [54]. Saturation was also explicitly defined as the point at which no new themes emerged after three consecutive interviews for each subgroup (ED men, ED women, and RHC women). To account for the iterative nature of qualitative research, the team regularly monitored emerging themes throughout data collection and concluded after 19 interviews when saturation was reached across all subgroups. This approach helped balance initial pragmatic estimates of sample size with the flexibility needed to capture all relevant themes.

**Sampling technique.** A purposive sampling strategy was employed to ensure representation across key demographic and experiential characteristics from the 655 participants who completed the initial quantitative survey. Participants were purposively selected to ensure diversity across key factors, including age, marital status, education level, occupation, tribe, and religion. Additionally, IDI participants were chosen to reflect varying personal experiences with and perspectives on alcohol use, from patients who only reported neutral or positive past experiences with alcohol, used to drink but are now abstinent, have a close friend or relative that drinks heavily, or suffered significant negative consequences from alcohol. Preliminary quantitative findings also informed participant selection to better explore notable patterns identified in the data. For example, early quantitative analysis indicated that divorced or widowed women had above-average alcohol intake, prompting the inclusion of a recently divorced woman with high alcohol consumption to explore this trend in greater depth.

To ensure a representative sample and minimize potential bias, the characteristics of IDI participants were reviewed monthly by the study lead. Any imbalances or gaps in representation were addressed through adjustments to subsequent participant selection. This iterative process ensured that diverse demographic and experiential perspectives were captured while aligning with the study's overarching focus.

**Data collection and instruments.** Qualitative data collection began approximately one month after initial survey collection in alignment with our sequential, explanatory study approach. During their initial quantitative survey, if research assistants identified participants perceived to be excellent candidates for IDI based on the previously described selection parameters, these patients were approached for IDI participation at the conclusion of their initial survey or in a later phone call. If expressing interest in participating, the same research assistant who had completed their quantitative survey with them set up a time to meet and conducted the interview in a private room at KCMC. All interviews were conducted in Kiswahili using a semi-structured interview guide with open-ended questions and probes. Research assistants were encouraged to ask their own clarifying questions when the information participants provided was unclear, conflicted with previous statements, or warranted further explanation.

The interview guide was piloted with the study lead and three KiSwahili-speaking research team members. These team members, who were not part of the final study sample, had professional backgrounds in nursing, medicine, and qualitative research, and were familiar with the target population and healthcare setting. Their feedback informed refinements to ensure the cultural and linguistic validity of the guide prior to initiating interviews with participants. Pilot interviews were conducted solely for refinement purposes and were not included in the study dataset. With the knowledge and consent of the participant, all IDIs were audio recorded using a handheld recording device for later transcription and translation, with a snack and drink provided midway through the interview. Participants were informed prior to participation that they would be given a small stipend (5000 TSH, equivalent to approximately $2 USD) to reimburse their travel expenses.

**Data analysis procedure.** A grounded theory approach was chosen as this topic remains poorly understood in the local context, with little prior research foundation from which to draw. Unlike thematic analysis, which is often used to organize and interpret pre-existing frameworks or themes, grounded theory is designed to generate new, theory-driven insights directly from the data itself [55–57]. This approach allowed us to iteratively learn from participants' responses and develop a holistic understanding of alcohol use perceptions specific to this setting, rooted in the data rather than external frameworks.

US and Tanzanian study team members co-developed the initial codebook from the first several IDIs. This codebook was inductively and deductively generated, using broad categories from the interview guide as an initial framework while also enabling organic generation of codes without preconceived notions, in alignment with grounded theory methodology [55–58]. The evolving codebook facilitated the creation of content memos for each emerging theme and sub-theme, which formed the basis of our qualitative findings.

Interviews were coded by both Tanzanian and US research team members who independently coded initial interviews and then met to discuss and resolve any coding discrepancies. This process was repeated until an 80% agreement was

reached. At that point, with approval from the research team, the lead analyst completed the coding of all remaining interviews, updating the codebook accordingly. After the preliminary quantitative analyses, qualitative and quantitative data collection occurred in parallel for the remainder of the study period. All qualitative analysis was conducted in NVivo software version 12 and approved by the entire research team prior to reporting these results.

### Ethics statement

Ethical approval for this study was obtained from the Duke University Institutional Review Board, the Kilimanjaro Christian Medical University College Ethical Review Board, and the Tanzanian National Institute of Medical Research before data collection. Personal health information was employed in screening and enrollment procedures, but was de-identified when collected, stored, and analyzed and shared only via a data share agreement.

## Results

### Quantitative

**Demographics.** Six-hundred and seventy-six patients started, and 655 patients completed the survey questionnaire. The average age of all ED and RHC participants was 41.7 years old (SD = 18.7) (Table 2). Most participants identified as Christian (80%), married (50%), and employed (57%) at the time of survey delivery. The income distribution varied, with 32% earning ≤ 50,000 TZS per month and 39% earning > 200,000 TZS per month. Education levels ranged widely, with 29% reporting college-level attainment and 8.7% reporting no formal education. Further details on demographics, including tribal affiliation, are reported in Table 2.

**Alcohol consumptions and related consequences.** Descriptive analyses revealed significant variation in alcohol consumption and related consequences across populations. ED men had the highest average AUDIT scores (mean = 6.76, SD = 8.16), followed by ED women (mean = 3.07, SD = 4.76) and RHC women (mean = 1.86, SD = 3.46). Similarly, DrInC scores showed that ED men experienced the most alcohol-related consequences (mean = 16.4, SD = 19.6), followed by ED women (mean = 9.11, SD = 13.1) and RHC women (mean = 5.47, SD = 9.33) (Table 3). DrInC subscores for physical, interpersonal, social responsibility, and impulse control consequences followed the same trend, where ED men followed by ED women and RHC women faced a higher burden of alcohol-induced physical, interpersonal, intrapersonal, impulse control, and social responsibility harms (Table 3, Fig 1). Of note, ED women showed the largest variability in DrInC scores in several subcategories, including physical and intrapersonal consequences (Fig 1).

Linear regression analyses demonstrated a significant linear association between AUDIT and DrInC scores across all groups (Fig 2). The association was strongest for ED women ($R^2 = 0.76$, $p < 0.001$), reflecting that higher alcohol use strongly correlates with more alcohol-related consequences in this group. The association was slightly weaker for ED men ($R^2 = 0.65$, $p < 0.001$) and RHC women ($R^2 = 0.53$, $p < 0.001$), indicating population-specific differences in the impact of alcohol consumption (Fig 2).

### Qualitative

Nineteen individuals (RHC women, n = 5; ED women, n = 5; ED men, n = 9) participated in IDIs. The ages of IDI participants spanned from 20 to 70 years, with a mix of education, income, and alcohol intake levels as illustrated in Table 4. Approximately a third of participants held positive views or remarked that they generally benefited from alcohol, almost half said these views and experiences were overall negative, and the remaining quarter of participants stated that it depended. Two main themes and several sub-themes related to community perceptions and the impact of alcohol emerged from the IDIs. The two main themes were Alcohol's perceived harms and Alcohol's perceived benefits (Table 5). Participants noted the ways they felt alcohol harmed either them or their community, which included a failure to fulfill their responsibilities, stigma, sexual assault, increased conflict, chronic health issues, and financial instability. On the other

**Table 2. Study Population Demographics.**

| Demographics by Population Type | Overall, N = 655[1] | ED Women, N = 271[1] | RHC Women, N = 270[1] | ED Men, N = 114[1] |
|---|---|---|---|---|
| **Age Category** | | | | |
| 18–24 | 110 (18%) | 47 (19%) | 53 (21%) | 10 (9.5%) |
| 25–34 | 159 (26%) | 41 (16%) | 92 (37%) | 26 (25%) |
| 35–44 | 106 (17%) | 42 (17%) | 45 (18%) | 19 (18%) |
| 44–54 | 94 (16%) | 47 (19%) | 32 (13%) | 15 (14%) |
| Over 55 | 137 (23%) | 75 (30%) | 27 (11%) | 35 (33%) |
| Missing/refused per block | 49/ 655 | 19/ 271 | 21/ 270 | 9/ 114 |
| **Average Age** | 41.7 (18.7) | 46.1 (22.8) | 36.4 (12.7) | 45.2 (16.4) |
| **Personal Income Category (TZS per month)** | | | | |
| 0 to 50,000 | 205 (32%) | 110 (41%) | 68 (25%) | 27 (25%) |
| 50,001–100,000 | 44 (6.8% | 17 (6.3%) | 15 (5.6%) | 12 (11%) |
| 100,001–150,000 | 56 (8.7%) | 17 (6.3%) | 29 (11%) | 10 (9.4%) |
| 150,001–200,000 | 91 (14%) | 34 (13%) | 37 (14%) | 20 (19%) |
| > 200,000 | 248 (39%) | 92 (34%) | 119 (44%) | 37 (35%) |
| Missing/refused per block | 11/ 655 | 1/ 271 | 2/ 270 | 8/ 114 |
| **Religion** | | | | |
| None | 11 (1.7%) | 5 (1.8%) | 3 (1.1%) | 3 (2.6%) |
| Christian | 522 (80%) | 218 (80%) | 222 (82%) | 82 (72%) |
| Muslim | 121 (18%) | 47 (17%) | 45 (17%) | 29 (25%) |
| Other | 1 (0.2%) | 1 (0.4%) | 0 (0%) | 0 (0%) |
| Missing/refused per block | 0/ 655 | 0/ 271 | 0/ 270 | 0/ 114 |
| **Highest Educational Attainment** | | | | |
| None | 52 (8.7%) | 30 (12%) | 7 (2.9%) | 15 (13%) |
| Primary | 182 (31%) | 82 (34%) | 61 (25%) | 39 (35%) |
| Secondary | 133 (22%) | 48 (20%) | 64 (27%) | 21 (19%) |
| College | 170 (29%) | 65 (27%) | 87 (36%) | 18 (16%) |
| Graduate | 13 (2.2%) | 4 (1.7%) | 2 (0.8%) | 7 (6.2%) |
| Vocational | 45 (7.6%) | 13 (5.4%) | 20 (8.3%) | 12 (11%) |
| Missing/refused | 60/ 655 | 29/ 271 | 29/ 270 | 2/ 112 |
| **Marital Status** | | | | |
| Never Married or Single | 135 (21%) | 55 (20%) | 56 (21%) | 24 (21%) |
| Living with a partner, not in a registered marriage | 79 (12%) | 26 (9.6%) | 41 (15%) | 12 (11%) |
| Living with a partner in a registered marriage | 327 (50%) | 128 (47%) | 140 (52%) | 59 (52%) |
| Divorced or Separated | 3 (4.7%) | 16 (5.9%) | 6 (2.2%) | 9 (8.0%) |
| Widowed | 81 (12%) | 46 (17%) | 27 (10%) | 8 (7.1%) |
| Missing/refused | 2/ 655 | 0/ 271 | 0/ 270 | 2/ 114 |
| **Employment Status** | | | | |
| Employed | 371 (57%) | 127 (47%) | 192 (71%) | 52 (46%) |
| Unemployed | 215 (33%) | 111 (41%) | 50 (19%) | 54 (47%) |
| Student | 69 (11%) | 33 (12%) | 28 (10%) | 8 (7.0%) |
| Missing/refused | 0/ 655 | 0/ 271 | 0/ 270 | 0/ 114 |
| **Tribe** | | | | |
| Chagga | 329 (50%) | 126 (46%) | 146 (54%) | 57 (50%) |
| Iraq | 21 (3.2%) | 9 (3.3%) | 8 (3.0%) | 4 (3.5%) |

*(Continued)*

**Table 2.** (Continued)

| Demographics by Population Type | Overall, N = 655[1] | ED Women, N = 271[1] | RHC Women, N = 270[1] | ED Men, N = 114[1] |
|---|---|---|---|---|
| Maasai | 25 (3.8%) | 13 (4.8%) | 7 (2.6%) | 5 (4.4%) |
| Mmeru | 29 (4.4%) | 14 (5.2%) | 11 (4.1%) | 4 (3.5%) |
| Muha or Non-African | 6 (1.0%) | 3 (1.1%) | 3 (1.1%) | 0 (0%) |
| Nyaturu | 17 (2.6%) | 7 (2.6%) | 10 (3.7%) | 0 (0%) |
| Pare | 70 (11%) | 27 (10.0%) | 25 (9.3%) | 18 (16%) |
| Sambaa | 26 (4.0%) | 7 (2.6%) | 10 (3.7%) | 9 (7.9%) |
| Sukuma | 34 (5.2%) | 10 (3.7%) | 16 (5.9%) | 8 (7.0%) |
| Other African | 98 (15%) | 55 (20%) | 34 (13%) | 9 (7.9%) |
| Missing/refused | 0/ 655 | 0/ 271 | 0/ 270 | 0/ 114 |

[1]n/ N (%); Mean (SD).

**Table 3.** Reported Average AUDIT and DrInC Scores across Patient Populations.

| | RHC Women (n = 270) | ED Women (n = 271) | ED Men (n = 114) | Total Population (n = 655) |
|---|---|---|---|---|
| **Overall AUDIT Score**, *missing: 0* | 1.86 (3.46) | 3.07 (4.76) | 6.76 (8.16) | 3.22 (5.36) |
| **Overall DrInC Score**, *missing: 17* | 5.47 (9.33, 75) | 9.11 (13.1, 80) | 16.4 (19.6, 88) | 9.00 (13.9,) |
| **Physical Subscore**, *missing: 17* | 0.60 (1.73, 12) | 1.39 (2.75, 15) | 2.54 (3.37, 12) | 1.28 (2.62, 15) |
| **Intrapersonal Subscore**, *missing: 17* | 0.54 (1.87, 14) | 1.00 (2.34, 20) | 2.86 (3.97, 17) | 1.15 (2.68, 20) |
| **Social Responsibility Subscore**, *missing: 17* | 0.52 (1.74, 13) | 1.15 (2.53, 14) | 2.53 (3.77, 17) | 1.15 (2.66, 17) |
| **Interpersonal Subscore**, *missing: 17* | 0.39 (1.51, 14) | 0.99 (2.67, 23) | 3.02 (4.42, 17) | 1.12 (2.91, 23) |
| **Impulse Control Subscore**, *missing: 17* | 0.34 (1.35, 17) | 0.70 (1.89, 13) | 1.99 (3.34, 17) | 0.80 (2.17, 17) |

Data are presented as mean (SD, range) for quantitative variables, and count (%) for qualitative variables.

hand, examples of how alcohol was viewed as beneficial within the community were mentioned, including fostering economic growth, creating an outlet to uphold cultural traditions, and instilling more social unity.

To preface this data, the communities' perceptions and expectations of alcohol use explored in this results section is the amalgamation of general themes and trends from 19 participants. While these findings have been grouped for clarity, it is important to note that there was a significant amount of variability in responses and opinions, especially across gender, "*various religions, clans, and tribes" (IDI #3, Female)*. As one participant explains:

> "*The community is the combination of many people with different ideologies about life; there is no way that my perception will be the same with everybody in the community. It's because we have been born and raised in different families and environments and everyone came up with his/her own life basics, so according to how one came across alcohol it is how the perception will be built." (IDI #19, Female)*

## Alcohol's perceived harms

As part of their interview, IDI participants were asked how alcohol use negatively impacted their community. Respondents stated that alcohol leads to stigma, sexual violence, risky sexual behaviors, and an inability of community members to

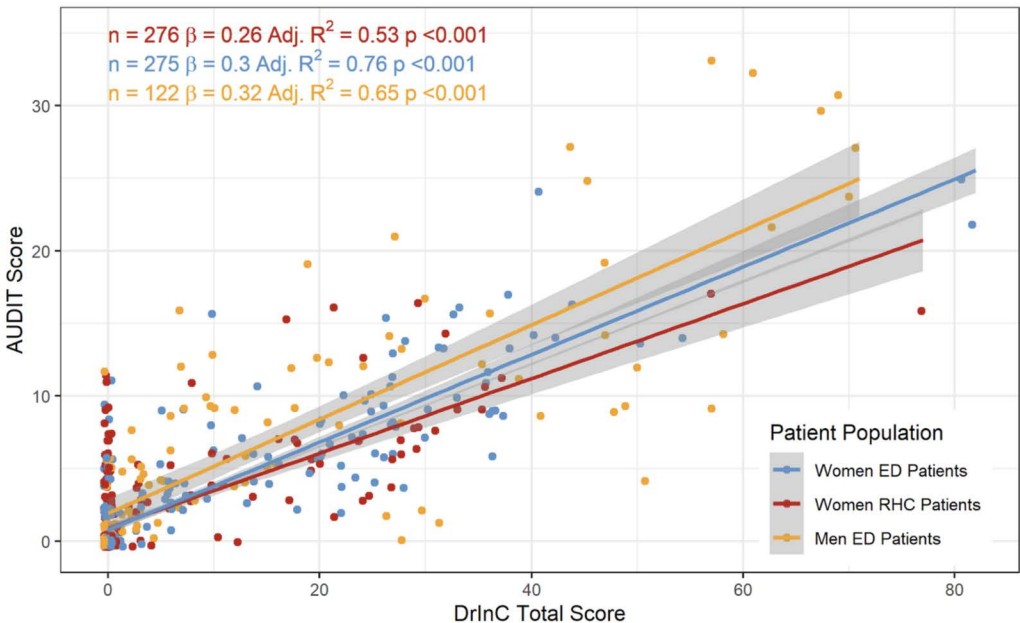

**Fig 1. Box-Plot Distribution of DrInC Scores and Subscores across Patient Populations.**

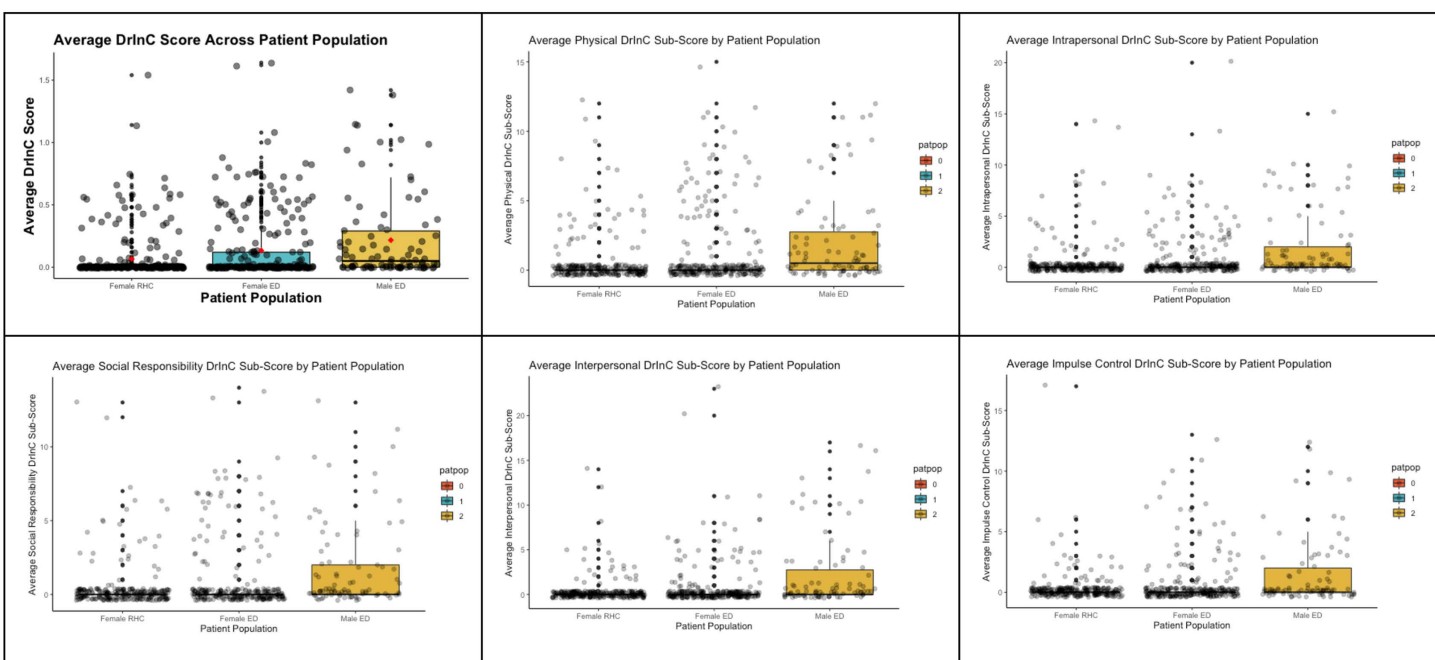

**Fig 2. Association of AUDIT and DrInC Scores across Patient Populations.**

**Table 4. Interviewee Demographics.**

| Demographics by Population Type | Overall, N = 19[1] | ED Women, N = 5[1] | RHC Women, N = 5[1] | ED Men, N = 9[1] |
|---|---|---|---|---|
| **Age Category** | | | | |
| 18–24 | 4/ 19 (21.1%) | 2/ 5 (40%) | 2/ 5 (40%) | 0/ 9 (0%) |
| 25–34 | 4/ 19 (21.1%) | 1/ 5 (20%) | 0/ 5 (0%) | 3/ 9 (33.3%) |
| 35–44 | 5/ 19 (26.3%) | 1/ 5 (20%) | 2/ 5 (40%) | 2/ 9 (22.2%) |
| 44–54 | 4/ 19 (21.1%) | 1/ 5 (20%) | 1/ 5 (20%) | 2/ 9 (22.2%) |
| Over 55 | 2/ 19 (10.5%) | 0/ 5 (0%) | 0/ 5 (0%) | 2/ 9 (22.2%) |
| **Personal Income (TZS), *missing: 7*** | 282500 (422807.18) | 360000 (505129.05) | 268000 (422807.18) | 125000 (106066.02) |
| **Household Income (TZS), *missing: 7*** | 516666.67 (484455.33) | 730000 (582329.07) | 390000 (484455.33) | 300000 (141421.356) |
| **Religion** | | | | |
| Christian | 15/ 19 (78.9%) | 4/ 5 (80%) | 5/ 5 (100%) | 6/ 9 (66.7%) |
| Muslim | 4/ 19 (21.1%) | 1/ 5 (20%) | 0/ 5 (0%) | 3/ 9 (33.3%) |
| **Highest Educational Attainment, *missing: 2*** | | | | |
| College | 9/ 19 (47.4%) | 3/ 5 (60%) | 3/ 5 (60%) | 3/ 9 (33.3%) |
| Graduate | 1/ 19 (5.3%) | 0/ 5 (0%) | 0/ 5 (0%) | 1/ 9 (11.1%) |
| None | 0/ 19 (0%) | 0/ 5 (0%) | 0/ 5 (0%) | 0/ 9 (0%) |
| Primary | 5/ 19 (26.3%) | 1/ 5 (20%) | 1/ 5 (20%) | 3/ 9 (33.3%) |
| Secondary | 2/ 19 (10.5%) | 1/ 5 (20%) | 1/ 5 (20%) | 0/ 9 (0%) |
| Vocational | 0/ 19 (0%) | 0/ 5 (0%) | 0/ 5 (0%) | 0/ 9 (0%) |
| **Marital Status** | | | | |
| Divorced or Separated | 2/ 19 (10.5%) | 1/ 5 (20%) | 0/ 5 (0%) | 1/ 9 (11.1%) |
| Living together with a partner but not in a registered marriage | 2/ 19 (10.5%) | 0/ 5 (0%) | 1/ 5 (20%) | 1/ 9 (11.1%) |
| Living together with a partner in a registered marriage | 9/ 19 (47.4%) | 2/ 5 (40%) | 2/ 5 (40%) | 5/ 9 (55.6%) |
| Never Married or Single | 5/ 19 (26.3%) | 2/ 5 (40%) | 2/ 5 (40%) | 1/ 9 (11.1%) |
| Widowed | 1/ 19 (5.3%) | 0/ 5 (0%) | 0/ 5 (0%) | 1/ 9 (11.1%) |
| **Employment Status** | | | | |
| Employed | 12/ 19 (63.2) | 3/ 5 (60%) | 5/ 5 (100%) | 4/ 9 (44.4%) |
| Unemployed & not a Student | 4/ 19 (21.1%) | 0/ 5 (0%) | 0/ 5 (0%) | 4/ 9 (44.4%) |
| Unemployed but Student | 3/ 19 (15.8%) | 2/ 5 (40%) | 0/ 5 (0%) | 1/ 9 (11.1%) |
| **Tribe** | | | | |
| Chagga | 9/ 19 (47.4%) | 2/ 5 (40%) | 4/ 5 (80%) | 3/ 9 (33.3%) |
| Iraq | 1/ 19 (5.3%) | 0/ 5 (0%) | 0/ 5 (0%) | 1/ 9 (11.1%) |
| Maasai | 1/ 19 (5.3%) | 0/ 5 (0%) | 0/ 5 (0%) | 1/ 9 (11.1%) |
| Mmeru | 2/ 19 (10.5%) | 0/ 5 (0%) | 0/ 5 (0%) | 2/ 9 (22.2%) |
| Nyaturu | 1/ 19 (5.3%) | 1/ 5 (20%) | 0/ 5 (0%) | 0/ 9 (0%) |
| Other African | 3/ 19 (15.8) | 2/ 5 (40%) | 1/ 5 (20%) | 0/ 9 (0%) |
| Pare | 1/ 19 (5.3%) | 0/ 5 (0%) | 0/ 5 (0%) | 1/ 9 (11.1%) |
| Sambaa | 1/ 19 (5.3%) | 0/ 5 (0%) | 0/ 5 (0%) | 1/ 9 (11.1%) |
| **Drinking Frequency** | | | | |
| 0 times/week | 5/ 19 (26.3%) | 1/ 5 (20%) | 1/ 5 (20%) | 3/ 9 (33.3%) |
| 1-2 times/week | 6/ 19 (31.6%) | 1/ 5 (20%) | 1/ 5 (20%) | 4/ 9 (44.4%) |
| 3-4 times/week | 7/19 (36.8%) | 2/ 5 (40%) | 3/ 5 (60%) | 2/ 9 (22.2%) |
| 5-6 times/week | 1/ 19 (5.3%) | 0/ 5 (0%) | 0/ 5 (0%) | 1/ 9 (11.1%) |
| Every day | 0/ 19 (0%) | 0/ 5 (0%) | 0/ 5 (0%) | 0/ 9 (0%) |
| Multiple times a day | 0/ 19 (0%) | 0/ 5 (0%) | 0/ 5 (0%) | 0/ 9 (0%) |

*(Continued)*

**Table 4.** (Continued)

| Demographics by Population Type | Overall, N = 19[1] | ED Women, N = 5[1] | RHC Women, N = 5[1] | ED Men, N = 9[1] |
|---|---|---|---|---|
| **Drinking Quantity** | | | | |
| >6 bottles | 0/ 19 (0%) | 0/ 5 (0%) | 0/ 5 (0%) | 0/ 9 (0%) |
| 0 drinks | 4/ 19 (21.1%) | 1/ 5 (20%) | 1/ 5 (20%) | 2/ 9 (22.2%) |
| 1-2 bottles | 8/ 19 (42.1%) | 2/ 5 (40%) | 3/ 5 (60%) | 3/ 9 (33.3%) |
| 3-4 bottles | 6/ 19 (31.6%) | 2/ 5 (40%) | 1/ 5 (20%) | 3/ 9 (33.3%) |
| 5-6 bottles | 0/ 19 (0%) | 0/ 5 (0%) | 0/ 5 (0%) | 0/ 9 (0%) |
| >6 bottles | 0/ 19 (0%) | 0/ 5 (0%) | 0/ 5 (0%) | 0/ 9 (0%) |
| Refused/Do not know | 1/ 19 (5.3%) | 0/ 5 (0%) | 0/ 5 (0%) | 1/ 9 (11.1%) |

[1]n/ N (%); Mean (SD).

**Table 5. Qualitative Themes and Sub-Themes.**

| Themes | Sub-Themes |
|---|---|
| Alcohol's Perceived Harms | Inability to Fulfill Familial Responsibilities<br>Stigma and Discrimination<br>Sexual Consequences<br>Interpersonal Conflict and Violence<br>Injuries and Chronic Disease<br>Financial Instability |
| Alcohol's Perceived Benefits | Economic Growth<br>Upholding Cultural Practices<br>Social Unity |

uphold individual responsibilities. Most also saw it as an instigator of physical, verbal, and emotional conflict among family members, and believed it contributed to long-term health issues and financial instability.

*"What I can say is the society do not have positive expectation about alcohol use, because we see families dis-integrated, people fall sick as a result of alcohol and many people at first drink but are able to work and carry on their daily activities and routine but later on they fail to carry on tasks because alcohol makes them weak." (IDI #3, Female)*

**Inability to fulfill family responsibilities.** Most participants mentioned that alcohol was an inhibitor in one's ability to fulfill their responsibilities to their family. Those who drank in excess would "*forget [their] responsibilities*" (IDI #11, Male), become too "*weak*" (IDI #3, Female), "*lazy*" (IDI #19, Female), or simply, "*fail to perform their daily duties*" (IDI #6, Female). One participant elaborates on their own experience:

*"I see alcohol is not such a good thing…It can make one forgetting everything about his/her family, even children can go to bed hungry without you having an idea because you're busy drinking. Children left home by drunkard parents can even experience assaults of different kinds just because the parents are not there to protect them…I advice my fellow parents and those who have families to care for to stop taking alcohol"* (IDI #7, Female)

**Stigma and discrimination.** All interviewees noted alcohol-related stigma to be present to some extent in their communities, but to varying degrees and concentrated in certain circumstances. One participant reported that they

"*haven't seen any stigma among men or women…unless if after drinking you cause problems like fighting or abusing other people*" (IDI#12, Male). As another explains, stigma could also arise when one fails to accomplish preset responsibilities or drinks unsafe alcoholic beverages:

> "*Alcohol intake in Tanzania is like a ritual. Anyone can drink. The stigma arises when someone fails to control their drinking habit, like when someone drinks all the time and fails to fulfill his or her obligations, or if someone drinks those homemade beers (local brews) that are not tested and not standardized by the bureau of standards. People who drink Gongo and Dadii are really stigmatized in the community. Because once you drink those, you may even fail to control your bowels, which leads to shame and embarrassment*" (IDI #2, Male).

In addition to the quality of liquor, another notable facet of alcohol-related stigma is its interaction with gender. Participants noted that "*women who drink are stigmatized more than men*" (IDI #7, Female) and that their "*community strongly hate women who drinks alcohol*" (IDI #19, Male). This arose in part because of aforementioned gender roles, where because "*women stay with the children most of the time*" (IDI #19, Male) there is concern that their alcohol habits will have greater negative repercussions on their children.

**Sexual consequences.** Interviewees reported that alcohol use caused or contributed to significant sexual consequences uniquely experienced by each gender. The first of these was the increased risk of acquiring a sexually transmitted disease – "*when you drink too much it can put you a risky of even practice unsafe sexual intercourse*" (IDI #4, Male), which generally stemmed from having more sexual partners:

> "*Men who drink lacks discipline, they don't respect themselves or their wives, they engage in sexual immorality without considering the fact that they are aged, married and with families.*" (IDI #7, Female)

A couple of men and women participants noted that alcohol could hinder their ability to perform sexual acts –

> "*A person cannot perform sexual activity well as before, to men they may encounter erectile dysfunction and women may also not be able to involve and perform sexual activity properly. Alcohol can make a person really desire the sexual intimacy but just not able to perform it because of the effect of alcohol in the brain.*" (IDI #16, Female)

This, though, was only mentioned by a few participants and stands in sharp contrast to the common theme of alcohol-related sexual assault. This final sexual consequence was described almost exclusively as men being violent against women – "*there is someone I know she was invited to a man house to drink alcohol it was her first time to drink alcohol she got drunk the man raped her*" (IDI #9, Female). Participants disagreed on whether or not men incurred sexual assault when drinking, one saying "*men…cannot experience sexual violence because it is not a common practice in Tanzania*" (IDI #2, Male), while another reported that "*men…when they are drunk, they have been sexually abused by the fellow men*" (IDI #13, Female).

**Interpersonal conflict and violence.** Sexual assault was not the only product of the increased violent and aggressive behavior that alcohol use incited. Interpersonal conflict and violence was a theme noted by three-quarters of participants. IDI #19 states,

> "*Alcohol causes misunderstandings and fights in society, I have witnessed many times that people get drunk and start to say abusive words, become very aggressive and arrogant.*" (IDI #19, Female)

Respondents described how alcohol use instigates emotional and physical conflict and distress within the family – "*I know many families that when the father is drunk, the mother and children become so anxious and afraid to sit with him closely. The most common violence is physical and emotional abuse*" (IDI #17, Male). As seen in this and the next quote,

aggression was described as being more common in men – *"it's very difficult to see women fighting after drinking. But men are always aggressive and violent"* (IDI #3, Female), with the ensuing conflict leading to family disintegration.

Moreover, several participants (4 of the 19 participants) discuss how a child's mental health is significantly affected by witnessing their parents *"fighting and shouting"* (INT#12, Male), a form of emotional conflict, due to alcohol consumption. As a result, many pre-existing interpersonal relationships were harmed or broken – *"I have witnessed many times that people get drunk and start to say abusive words to one another...I can say that alcohol can truly bring people together but can also destroy people's relationship"* (IDI #19, Female).

**Injuries and chronic disease.** While all the categories of harms thus far discussed can lead to significant physical, mental, and emotional problems, a plethora of other long-term health issues stemming from alcohol use like injuries and road traffic incidents, illness and chronic disease were also discussed by most participants.

> *"Alcohol can lead to organ damage like liver so there is a lot of complications...I had a friend of mine who was drinking every day, and he was not eating well so he ends up dying. Because of alcohol intoxication. But before that he was having anemia (low blood), low body weight, and muscle wasting"* (IDI #12, Male)

Health issues that were attributed to alcohol use by participants included the following: liver cirrhosis, cancer, kidney failure, *"puffy cheek"* (water retention resulting from alcohol use), neurological insults, and even death. Some participants even recount how local brews —which are *"not tested and not safe"* (IDI #4, Male) and have been characterized as *"strong alcohol"* (IDI #8, Male)— may incite even greater health consequences in regular consumers.

**Financial instability.** Lastly, almost half of the participants mentioned economic loss and financial instability, encouraged via a variety of different mechanisms, as a major source of personal and community harm. First, participants proposed that economic loss and financial instability seem to arise from the perceived difficulty of drinkers to find or maintain employment. That is, participants agreed that employed individuals who drink experience a loss of productivity. IDI #17, for example, stated, *"Alcohol-related behaviors may lower the individual performance at (the) workplace which leads to inadequate or inefficient production"* (IDI #17, Male).

In addition, if an individual spends *"a lot of money buying alcohol"* (IDI #12, Male) for themselves, this can take a toll on personal savings –*"I lost my business because of drinking too much"* (IDI #12, Male). One participant describes the monetary loss alcohol-related spending could cause:

> *" Alcohol affected my uncle so badly, he was so rich, and helped people in his family and relatives but once he became addicted with alcohol everything turns to zero. He lost everything, even closer friends. He is unable to take care of his family even paying school fees for his children!"* (IDI# 5, Male)

As men especially were seen to be the primary breadwinners for their families, a father's excess spending on alcohol could *"decrease in family income"* (IDI #17, Male). This created negative repercussions for a man's wife and children and permanently cripple their financial holdings:

> *"Alcohol is a reason for many families here to fall into extreme poverty because of men as family providers spend a lot of resources in alcohol while they just earn little to help their families survive as most of them work as drivers of tricycles and motorcycles. So, when a man has no control of his drinking habits and spends all of his income into drinking and definitely the family get into a vicious cycle of poverty."* (INT #19)

While increased alcohol consumption was seen as contributing to personal or familial financial distress for some, as will be discussed in the next section, several participants noted a financial benefit to increased alcohol use for sellers or the government.

**Alcohol's perceived benefits**

When asked how alcohol has positively impacted their community, a third (only one of which was female) commented on the economic gain in alcohol production and sales, and half of the participants commonly mentioned social and/or cultural benefits.

   **Economic growth.**  In contrast to the last sub-theme presented, IDI #2 described the monetary benefit alcohol brought to his community, saying it

   *"contributes to the economic growth of an individual who is selling, a family or a national income. The government collects taxes from the brewing industry, which helps to increase national income" (*IDI# 2, Male*).*

Interestingly though, while the government benefited financially from alcohol sales, several respondents discussed how *"local brew"* (IDI #4, Male) is often not effectively tested for alcohol content levels and results in harmful consequences for their community. One participant explained this lack of government monitoring:

   *"The community views [alcohol] as a burden promoted by the government, it is the government that legalized it and allow people to manufacture alcohol. And sometimes the government fails to monitor these industries and see what kind of alcohol they are producing. It is very unfortunate that some of the businessmen are not faithful and manufacture alcohol which are very strong and damage people's brain." (IDI #12, Male)*

Importantly, while the production of alcohol was seen to bring economic benefit for some, even this perceived boon carried with it the potential for harm to community members.

   **Upholding cultural practices.**  As mentioned in the first quote in this manuscript, alcohol use was tied to cultural and tribal events – *"Many people consider alcohol to be part of their social culture and way of life"* (IDI #2, Male). Some respondents found it appropriate to consume alcohol at cultural events or special occasions, using alcohol during rituals as a way to "*signify respect and honor to our ancestors*" (IDI #2, Male) and "*honor their ancestors' way of living, social customs that have been practiced for generations and generations*" (IDI #3, Female). One woman summarized these sentiments:

   *"There are social events that cannot be done without alcohol, for example, if you go to pay for dowry you have to bring alcohol to the event and even in some rituals, they must brew alcohol during the ritual service. So, there's a positive way alcohol affects our society as a whole…many people drink alcohol to spend time together and enjoy" (IDI #9, Female)*

   **Social unity.**  Finally, alcohol's positive influence on the social aspect of community life was a common theme in IDIs – *"alcohol helps people interact and have time to discuss different issues…it strengthens unity among people" (IDI #4, Male)* and

   *"[Alcohol] helps to bring people together in unity. You may find friends or just random people sitting together, making stories about a thing or two and celebrating their achievement over a glass of beer. In my society, you cannot just gather around grown-up people to discuss about something and just give them soda or juice, you must offer them alcohol so that they will listen to you properly"(IDI #19, Female).*

   *"The good results of alcohol, to me personally it has made my social life very great and I have been meeting a lot of people when I go for a drink, for my friends as we enjoy drinks together and talk about our lives but I have also met important and influential people to discuss important matters in business and work as we enjoy 1 or 2 glass of beer" (IDI #10, Female)*

Not only was alcohol use noted as bringing people together, but alcohol was also mentioned as having a key role in resolving social conflict and social gatherings:

> *"If you have misunderstandings with friends or family members, and you want to resolve those disputes…you need to offer them alcohol while you talk! You can't offer them soft drinks! Even when you want to get married, you need to buy a lot of alcohol for elderly people to drink and enjoy"* (IDI #2, Male).

## Discussion

This analysis is the first of its kind to provide a holistic, mixed-methods overview of the perceived harms and benefits of alcohol as perceived by ED and RHC patients in Moshi, Tanzania. Existing literature on Moshi has described the scope of alcohol use in this region [26,28,32,59], examined the determinants of use [60], and explored specific consequences related to alcohol, such as drunk driving [61] and stigma [29,33]. Perceptions of alcohol use have previously been explored [23], but without looking at alcohol-related consequences, and exclusively from the perspectives of injury patients. Our paper extends existing research in Moshi through prospectively collected surveys and IDIs to offer a more nuanced, comprehensive understanding of alcohol's social, economic, and physical implications among patients, their families, and community members. We used quantitative methods (DrInC scale) to identify the broad scope and frequency of alcohol-related consequences, while qualitative interviews added depth, revealing perceptions of these consequences and their sociocultural context. Our results show that individuals and their families experienced alcohol-related social harms like stigma and interpersonal conflict; however, alcohol was also thought to foster unity at the broader community level. From a social dimension, while patients perceived alcohol to have some level of financial benefit to the community and government through sales and taxes, substantial economic harm from excess spending on alcohol was noted to arise for the individuals and families of heavy users. Finally, no health benefits were linked to alcohol consumption, but a slew of physical harms were mentioned by participants.

In this analysis, we found that participants saw both positive and negative impacts of alcohol use at the individual, interpersonal, and community levels. While alcohol was viewed as contributing to harms like interpersonal violence, stigma, and financial instability, it was also recognized for its role in fostering social unity and maintaining traditional customs within Moshi. Participants frequently describe alcohol as a facilitator of unity, enabling individuals to come together for celebrations, rituals, or even conflict resolution. These findings align with global literature on the duality of alcohol's social effects, where its consumption is not only a public health challenge but also an entrenched part of social and cultural life [62–64]. One notable consideration of these findings is the discrepancies in gender. While male ED patients experienced the greatest variety in alcohol-related consequences, qualitatively, stigma was found to affect women more severely. For example, our quantitative analysis revealed that men responded affirmatively to experiencing approximately three different types of interpersonal consequences related to alcohol use via the DrInC scale, compared to women who experienced one or less. Comparatively, within IDIs, participants noted that women who drank were more harshly judged than men, with societal expectations of caregiving and child-rearing intensifying the perceived negative consequences of their alcohol use. These mixed-methods findings demonstrate that the social ramifications of alcohol use are significant yet nuanced, with men facing a broader range of social stipulations, while women endure disproportionately severe, stigmatizing consequences.

Previous Moshi-based studies have identified similar patterns in gender disparities in alcohol-related stigma and other alcohol-related harms like a willingness to engage in drunk driving [29,33,64,65]. Our findings build on this work by illustrating how alcohol's social benefits—such as creating opportunities for dialogue and strengthening communal ties—are deeply intertwined with its harms. Many participants noted alcohol's ability to generate conflict, going as far as to destroy romantic and family relationships. This underlies previous literature found alcohol to be a mediator in conflict and intimate

partner violence both globally and regionally [66–69]. These findings highlight the dual role of alcohol in Moshi, where it is seen to foster unity and traditions while also contributing to conflict, stigma, and gendered disparities.

Similar to social implications, alcohol consumption was found to have important financial consequences at the individual and country levels. IDI participants described financial strain from excessive alcohol-related spending, a finding that echoes previously published study data where nearly 40% of male ED patients spent between 10–46% of their average monthly household income on alcohol alone [42,70]. Our analysis highlights a gendered component to alcohol-related financial consequences as well, with men scoring over twice as high as women in the social responsibility category of drink DrInC, which encapsulates questions like 'I have missed days or work' and 'I have spent too much'. Other important economic-based findings from this analysis are the impacts at broader community and government levels, the effects of which were noted by participants to be positive. IDI participants believed that high rates of consumption were advantageous for sellers of alcohol, and rising national excise taxes on alcohol products contributed to increasing revenue for the government of Tanzania. However, McCoy et al. note that women who produce alcohol in Tanzania have limited control over any earned income and are placed at greater risk of sexual assault or unsafe sex practices, indicating that the benefits for sellers may exist primarily for men [71]. Furthermore, in our IDIs, specifics of how this additional revenue was used to benefit Tanzanians were not mentioned.

Currently, while there is a lack of literature outlining the broader financial implications of alcohol use within the East African context, Matzopoulos et al. estimate that within South Africa, the tangible and intangible costs incurred from harmful alcohol consumption comprise between 10–12% of the country's total GDP and outweigh the economic benefits associated with alcohol taxes and sales [72]. Further, research has suggested the inverse relationship between alcohol consumption and labor force participation globally [73]. This contributes to larger macroeconomic implications, including losses in economic productivity and development. Our data suggests that financial benefits from alcohol use were perceived to only exist for sellers of alcohol and at the government level, whereas for individuals and the broader community, participants almost exclusively noted the negative monetary impacts. Based on these results and the scarce regional data, we suggest that future research examine the micro- and macroeconomic implications of alcohol to inform policy recommendations in Tanzania.

Along with the social and financial consequences of alcohol consumption, we noted several physical harms stemming from alcohol use. Male ED patients were seen to experience a wider range of physical consequences than women presenting at the ED and RHC, as illustrated by our DrInC physical harm scores. Men answered positively to experiencing 2.54 alcohol-related physical harms compared to 0.60 from female RHC patients and 1.39 female ED patients. IDI data provided more information on the types of physical harm suffered by this patient population. These listed harms included increases in sexual assault, violence, risky sexual behaviors, and chronic health issues, with no mention of any physical benefit to themselves or their communities. This last finding echoes a recent statement made by the World Health Organization (WHO) stating that any level of drinking is harmful to physical health [74,75].

Participants' accounts of alcohol-related physical consequences align with findings from other studies in sub-Saharan Africa. Harmful alcohol use in the region has been linked to road traffic injuries, drunk driving, and other risky behaviors, with men experiencing higher rates of physical harm [32,61,76,77]. In addition, studies in Tanzania, Uganda, and South Africa have found an association between rape, sexual and intimate partner violence, and alcohol consumption [78–80]. Last, our IDI data also echoes the WHO and the Center for Disease Control's growing concern over the chronic health consequences of excess consumption, like liver and kidney failure [81]. Effective alcohol-reduction programs stand to reduce not only the negative health effects associated with alcohol but also violent behaviors that incite injuries. For example, Walton et al. and Ward et al. found that interventions aiming to lower alcohol intake subsequently reduce aggression [82,83]. Given the notable physical risk that alcohol poses, there is a need to address alcohol use behaviors in the Moshi area. Additionally, as research has shown that interventions are more effective when culturally adapted to specific populations [84,85], future alcohol-reduction-related programs and policies implemented in Moshi should encompass the specific socio-cultural nuances of the region as highlighted here.

By integrating quantitative and qualitative methods, this study captures both the measurable impacts of alcohol and the perceptions that shape its role in Moshi's sociocultural and economic fabric. In considering the social, financial, and physical impacts of alcohol use, one final note is that while broader community benefits like social unity, the upholding of traditions, and increased government revenue were raised, participants identified few personal benefits of alcohol use. This raises the question of whether alcohol truly has benefits in the community outside of historic incorporation into cultural and social events. Additional research exploring alcohol's impact at the personal versus community levels are thus recommended. Building on existing alcohol research in the region, insights from this analysis can help guide future culturally relevant research and alcohol reduction efforts in Moshi, and inform similar interventions in other communities.

## Recommendations

Based on the data we present, we recommend four key areas for future research, intervention, and policy efforts: 1) First, researchers should further explore the perceived social benefits associated with alcohol use, particularly its role in fostering and maintaining social unity within Moshi. This knowledge will facilitate a better understanding of the drivers of alcohol's current widespread use, even among harmful consequences. 2) To build upon further research and maximize program efficacy, alcohol reduction interventions should consider the distinct sociocultural contexts of alcohol use within this region and implement these nuanced considerations within intervention study designs. This will allow researchers and health professionals to create more culturally sensitive programs that can more effectively reduce local alcohol misuse.3) Finally, the micro- and macroeconomic implications of alcohol use, including perceived financial benefits and economic losses, should be analyzed to inform evidence-based alcohol-related policies in Tanzania.

## Limitations and directions for future research

Limitations for this analysis are present primarily within the quantitative data set and the DrInC tool. First, the rate of patients who screen failed or declined survey participation was not originally collected. This has an impact on the generalizability of our quantitative data to the ED and RHC patient populations at large. Additionally, this data was collected in the midst of the COVID-19 pandemic. As such, for the safety of our Tanzanian research staff, those who presented to the ED for COVID-19-related symptoms were not approached for study participation. While it was necessary to take these safety precautions, this also limited our ability to reach the full expected patient population. In both these cases, however, because of the large sample size and well-enforced sampling strategy, we still hold our data to be representative of the patient population. For DrInC, while this tool helps measure the range of consequences one might experience from alcohol use, it does not provide information on how severe these consequences are to the individual, which limits our understanding of alcohol's true impact. Finally, several important gender-specific consequences of alcohol use arose in this analysis, highlighting the need for further exploration of how alcohol differentially impacts men and women in this region. Even with these limitations, our paper offers one of the first published assessments of community perceptions of alcohol use in sub-Saharan Africa, offering insight into how alcohol is viewed both positively and negatively by male and female patients in Moshi, Tanzania. Our work highlights the need to consider social, financial, and cultural elements when addressing alcohol use disorders. Lastly, we recommend that local policy and healthcare efforts acknowledge and incorporate the roles of alcohol use in future alcohol-related programming and treatment initiatives in the Moshi community.

## Conclusion

Alcohol use in Moshi, Tanzania, was perceived to have both positive and negative impacts on patients and their communities. Social harms such as stigma, interpersonal conflict, and intimate partner violence were contrasted by perceived benefits like fostering social unity and upholding cultural traditions. Economic harms, including excessive household spending and decreased income, outweighed the broader financial benefits of alcohol sales and tax revenue. Physically, alcohol use was associated with injuries, risky sexual behaviors, and chronic health conditions, with no reported physical benefits. These findings offer critical insight into the perception of alcohol in Moshi as both a source of harm and a

facilitator of social cohesion, underscoring the need to integrate sociocultural, financial, and physical considerations into alcohol-reduction programming and policy efforts. Future research should examine alcohol's broader implications to guide effective, evidence-based policies and programs.

## Supporting information

**S1 Checklist. STROBE checklist.**
(DOCX)

**S2 Checklist. PLOS ONE clinical studies checklist.**
(DOCX)

**S3 Checklist. PLOS ONE human subjects research checklist.**
(DOCX)

**S1 Questionnaire. PLOS inclusivity in global research questionnaire.**
(DOCX)

## Author contributions

**Conceptualization:** Alena Pauley, Bariki Mchome, Blandina T. Mmbaga, Catherine A. Staton.

**Data curation:** Alena Pauley, Yvonne Sawe, Mariana Mikindo, Joseph Kilasara.

**Formal analysis:** Bariki Mchome, Blandina T. Mmbaga, Catherine A. Staton.

**Funding acquisition:** Alena Pauley, Catherine A. Staton.

**Investigation:** Bariki Mchome, Blandina T. Mmbaga, Catherine A. Staton.

**Methodology:** Alena Pauley, João Ricardo Nickenig Vissoci, Catherine A. Staton.

**Project administration:** Catherine A. Staton.

**Supervision:** Sharla Rent, Blandina T. Mmbaga, João Ricardo Nickenig Vissoci, Catherine A. Staton.

**Writing – original draft:** Alena Pauley, Madeline Metcalf, Mia Buono, Kirstin West, William Nkenguye.

**Writing – review & editing:** Alena Pauley, Madeline Metcalf, Mia Buono, Kirstin West, Sharla Rent, William Nkenguye, Catherine A. Staton.

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
