## [Decision Letter · Decision Letter 0]

5 Dec 2023

PONE-D-23-33073Understanding the Impacts and Perceptions of Alcohol Use in Northern Tanzania: A Mixed-Methods AnalysisPLOS ONE

Dear Dr. Staton,

Thank you for submitting your manuscript to PLOS ONE. After careful consideration, we feel that it has merit but does not fully meet PLOS ONE’s publication criteria as it currently stands. Therefore, we invite you to submit a revised version of the manuscript that addresses the points raised during the review process.

We look forward to receiving your revised manuscript.

Kind regards,

Fabiola Vincent Moshi

Academic Editor

PLOS ONE

Journal Requirements:

Additional Editor Comments:

Editor’s Comments

1. List the authors before the authors’ affiliation

2. Authors contribution has to come at the end

3. No need of repeating the title again in page 3

4. Have a heading abstract, then its sub-heading. Conclude the background of abstract with the general purpose of the study. In the method section have a sentence on the data collection method and tool, as well it is not clear on the data analysis method used in the study. Conclusion section is not talking on the key findings obtained from the quantitative part of the study, as well in the conclusion what do you mean be ED men and ED women? These were men and women attending emergency department they cannot be called ED men and women

5. In line 79, the sentence has language issue, line 112 change the sentence to past tense, this is a report

6. In the method section,

a. line 118, what do you mean by secondary analysis, did the study used secondary data? How while you have qualitative part of the study

b. What was the study design, mixed method is not a design rather an approach

c. Re-arrange this section to subheadings as follows

i. Study setting-justification of the selected setting and design, which design in the mixed method was used?

ii. Study population- the inclusion and exclusion criteria

iii. Quantitative part

1. Study design

2. Sample size estimation

3. Sampling technique

4. Data collection method and instrument

5. Variables and variables measurement

6. Data analysis procedure

iv. Qualitative

1. Study design

2. Sample size estimation

3. Sampling technique

4. Data collection method and instrument

5. Data analysis procedure

This is important for clarity, in the current state the section is not clear

d. Line 137 there is RHC, this is mentioned for the first time, please clarify

e. How did you conduct data analysis? What analysis software did you use?

7. In the results section

a. Describe the study respondents’ characteristics (frequency distribution table)

b. No need for table 3 after you have attended a above

8. In discussion section

a. No discussion for quantitative results

9. General questions

a. Why men and women in emergency department?

b. Why mixed method, what did you want to achieve by combining the approaches (quantitative and qualitative), at what point did these two approaches meet?

c. Why health facility study and not community-based study, what motivated you to select health facility and ED in particular?

Reviewers' comments:

Reviewer's Responses to Questions

**Comments to the Author**

1. Is the manuscript technically sound, and do the data support the conclusions?

Reviewer #1: Partly

Reviewer #2: Yes

2. Has the statistical analysis been performed appropriately and rigorously? 

Reviewer #1: No

Reviewer #2: N/A

3. Have the authors made all data underlying the findings in their manuscript fully available?

Reviewer #1: No

Reviewer #2: No

4. Is the manuscript presented in an intelligible fashion and written in standard English?

Reviewer #1: No

Reviewer #2: Yes

5. Review Comments to the Author

Reviewer #1: Introduction:

• Knowledge gap is not adequately described

Methods:

Study design

• Study design not clearly described. The author describe using secondary data analysis while on the other hand the authors describe the method for data collection were survey for quantitative data and in-depth interview for qualitative data. Is the author using already existing data for secondary analysis or collected the primary data during the study time?

Study sample

• Description regarding study sample/population not clear. The author does not explain the rationale of using female participants who attended RCH clinic while based on standard the pregnant woman or lactating mother were the mostly attending RCH clinic and are not allowing to take any kind of alcohol at that time due to pregnancy or lactation.

• Also the inclusion and exclusion criteria is not clearly stated

• The reasons for stratified sampling is not stated.

Data collection and Analysis

• Data collection is not clear. The reason of using systematic random sampling for male participants attending at Emergency Department and Female participant attending at RCH clinic only while including every female participant attending ED not stated

• It is also not stated how the data analysis would be adjusted to accommodate the stratification

• Data analysis for qualitative data is not clear stated. The author describes using thematic analysis and grounded theory. Which method is used in analyzing qualitative data?

• The reasons of using grounded theory is not stated, also, it is not utilized either.

• To report if the questionnaire have been tested for internal validity not stated

Results

• The author did not describe the demographic characteristics of the participants participating in quantitative survey

• Most of the quantitative data before regression analysis is not supported by statistical data

• Data analysis in qualitative data is not sufficiently described. How does the end up with themes and sub themes without codes and emerging themes?

Generally

• I think this is a good study with social relevance

• The protocol is in Standard English; however, there are numerous typographical and grammatical errors.

• I think the manuscript need major revision before considered for publication.

• I think the author needs to provide the recommendations to relevant authority based on the findings of the study.

Reviewer #2: Generally, the manuscript is well written and the authors tried to capture every issues related to alcohol use while reflecting the community as a victim, however there are several observations noted that need to be addressed.

L114: The line sate that "Factors underlying differential expectations of use will also be explored" how was this included in this study?

L137: Were all the participants recruited involved with alcohol use (already in use of alcohol) or anyone who falls in an interval of 3 in either EMD or RCH was involved?

L152: The line is openly stated but I suggest it to be restated as "researcher administered questionnaire was employed so as …"

L163: Although there are no clear cutoff points for DrInC tool, what were the cutoff points used by previous studies that could be used as reference? OR what was the reference used to determine the cutoff points used in the current study?

L202: Was there any software tool used to analyze the qualitative data?

DISCCUSSION PART:

The discussion part is good, however the perceived harm which is one of the qualitative results was more discussed as compared to perceived benefits. the comparison from different study could provide more informations which can be used for future interventions of alcohol use disorders. Also, would suggest to add different studies which did not concur with the results obtained from the current study so as to bring more challenges to the future researchers.

6. PLOS authors have the option to publish the peer review history of their article (what does this mean? ). If published, this will include your full peer review and any attached files.

**Do you want your identity to be public for this peer review?** For information about this choice, including consent withdrawal, please see our Privacy Policy .

Reviewer #1: No

Reviewer #2: No

---

## [Author Response · Author response to Decision Letter 1]

19 Feb 2024

February 9th, 2024

To Whom It May Concern:

We appreciate the helpful suggestions presented to us by our reviewers and thank them for the opportunity to improve our manuscript. We have incorporated the changes requested by both reviewers and our academic editor in our attached submission with our specific modifications outlined below. Our revised manuscript has been attached to this resubmission packet.

Reviewer #1:

Comment: Knowledge gap is not adequately described.

Response: We thank the reviewer for this comment. We have edited the final paragraph of the introduction to more clearly define the knowledge gap that we aimed to address with this research. This modification now reads, ‘However, literature has yet to explore the specific societal perceptions and implications of alcohol use within the Moshi community. Understanding the full scope of alcohol’s impact and why its use remains endemic is essential to mitigating its effects. This knowledge, in turn, can be used to better inform and shape future alcohol-reduction interventions.’

Comment: Study design not clearly described. The authors describe using secondary data analysis while on the other hand the authors describe the method for data collection were survey for quantitative data and in-depth interview for qualitative data. Is the author using already existing data for secondary analysis or collected the primary data during the study time?

Response: We thank the reviewer for this clarification. We have updated the study design section of our manuscript to read, ‘This study used a sequential explanatory mixed-methods approach to explore community perceptions and implications of alcohol use among KCMC’s Emergency Department (ED) and Reproductive Health Center (RHC) patients in Moshi, Tanzania.’ In this revision, we have removed mention of a secondary analysis as primary data was collected during the study timeline and is analyzed in this manuscript.

Comment: Description regarding study sample/population not clear. The author does not explain the rationale of using female participants who attended RCH clinic while based on standard the pregnant woman or lactating mother were the mostly attending RCH clinic and are not allowed to take any kind of alcohol at that time due to pregnancy or lactation.

Response: We appreciate this feedback from the reviewer and the opportunity to improve our manuscript in addressing it. We have added greater detail around our decision to enroll RHC patients under the section ‘Study Setting.’ This addition reads, ‘the RHC…was selected because of its primarily female patient population, which facilitated a richer, gynocentric perspective on alcohol use which has been under-represented in past local research. Measuring alcohol use in pregnant or postpartum women was also important for maternal and child health given the high rates of alcohol use during pregnancy previously described in this setting.’ The data we collected on alcohol use in pregnancy will be explored in an upcoming manuscript from our team.

Comment: The inclusion and exclusion criteria is not clearly stated.

Response: We appreciate this feedback from the reviewer. We have more clearly stated our eligibility criteria in our updated section entitled ‘Study Population.’ This addition reads, ‘Those enrolled met the following eligibility criteria: 1) fluency in Kiswahili, 2) ability to provide informed consent, 3) aged 18 or older, 4) not a prisoner, and 5) received initial care at KCMC’s ED or RHC.’

Comment: The reasons for stratified sampling is not stated.

Response: We appreciate this comment from the reviewer. We have updated the “Study Design and Sampling Technique” subsection of the methods to outline the enrollment groups and rationale in accordance with our study aims. This addition reads, ‘a systematic random sampling strategy was employed to acquire a representative sample of our three patient populations, male ED, female ED patients, and female RHC patients, and better describe the characteristics of each. Except for ED females, every third patient was approached for potential using the ED triage or RHC intake registries. To meet enrollment goals given that significantly fewer women seek care at the ED than men, every eligible female ED patient was approached for study participation’

Comment: Data collection is not clear. The reason of using systematic random sampling for male participants attending at Emergency Department and Female participant attending at RCH clinic only while including every female participant attending ED not stated

Response: We appreciate this feedback from the reviewer and have updated the relevant section as such. Under the same section specified in our comment above, our manuscript now reads: ‘Except for ED females, every third patient was approached for potential study participation using the ED triage or RHC intake registries. For women seeking ED care, every eligible individual was approached for study participation. This was done as significantly fewer women seek care at the ED and more women were needed in this particular study to accurately determine the prevalence of AUD.’

Comment: It is also not stated how the data analysis would be adjusted to accommodate the stratification

Response: We thank the reviewer for this comment. Given that our study sought to describe alcohol-related consequences and perceptions primarily as they relate to gender and clinical unit, data stratification was not performed. Instead, descriptive statistics were used in this analysis with the intention of understanding these populations holistically.

Comment: Data analysis for qualitative data is not clearly stated. The author describes using thematic analysis and grounded theory. Which method is used in analyzing qualitative data?

Response: We appreciate the reviewer for bringing this gap in our methods to our attention. We have heavily revised our qualitative data analysis section to list a grounded theory methodology as well as provided a more in-depth discussion of the exact analysis methods employed. This modification can be seen in our ‘Data Analysis Procedure’ subsection of our ‘Qualitative Data’ methodology.

Comment: The reasons for using grounded theory is not stated, also, it is not utilized either.

Response: We appreciate this comment from the reviewer. We have made substantial changes to our qualitative data analysis section to better describe why this methodology was chosen and how it was utilized in our analysis process.

Comment: To report if the questionnaire has been tested for internal validity not stated.

Response: We appreciate this feedback from the reviewer. We have updated our “Data Collection and Instruments” subsection of our quantitative methods to incorporate the previous cultural adaptation and validation of the DrInC instrument in this study setting. We have additionally modified our “Data Collection and Instruments” subsection of our qualitative methods to read, ‘the interview guide was piloted by research team members at KCMC to ensure interview relevancy and cultural and linguistic validity prior to the commencement of the interviews.’

Comment: The author did not describe the demographic characteristics of the participants participating in quantitative survey.

Response: We appreciate this feedback. We have included an additional table containing the demographic characteristics of the entire study sample in this revised manuscript version.

Comment: Most of the quantitative data before regression analysis is not supported by statistical data

Response: We thank the reviewer for bringing this feedback. We have added a table on study population demographics to the beginning of our quantitative results section.

Comment: Data analysis in qualitative data is not sufficiently described. How does the end up with themes and sub themes without codes and emerging themes?

Response: We appreciate this important question from the reviewer. We have updated our qualitative data analysis section to better describe our analysis process. The relevant part of this addition reads, ‘US and Tanzanian study team members co-developed the initial codebook from the first several IDIs. This codebook was inductively and deductively generated, using broad categories from the interview guide to provide a general framework for qualitative themes along with organic generation of codes without preconceived notions from the study team in concordance with grounded theory methodology. The themes established within the codebook facilitated the creation of content memos for each emerging theme and sub-theme which in turn provided the basis for our qualitative findings section below. The codebook remained an evolving document as new themes emerged throughout data collection.’

Comment: The protocol is in Standard English; however, there are numerous typographical and grammatical errors.

Response: We appreciate this feedback from the reviewer. We have revised this manuscript to address grammatical errors.

Comment: I think the author needs to provide the recommendations to relevant authorities based on the findings of the study.

Response: We appreciate this suggestion from the reviewer and have added a recommendation to the relevant authorities at the conclusion of our discussion section. This addition reads, ‘we recommend that local policy and healthcare efforts acknowledge and incorporate the roles of alcohol use in future alcohol-related programming and treatment initiatives in the Moshi community.’

Reviewer #2:

Comment: L114: The line states that "Factors underlying differential expectations of use will also be explored" how was this included in this study?

Response: We thank the reviewer for bringing this incongruity to our attention. As this manuscript did not fully explore factors underlying expectations of use, we have removed this sentence from the manuscript.

Comment: L137: Were all the participants recruited involved with alcohol use (already in use of alcohol) or anyone who falls in an interval of 3 in either EMD or RCH was involved?

Response: We thank the reviewer for this question. To answer your initial inquiry, perceived or measured alcohol use was not considered when assessing individuals for enrollment eligibility. We have updated the “Study Population” section of our methods to clarify this point accordingly.

Comment: L152: The line is openly stated but I suggest it to be restated as "researcher administered questionnaire was employed so as …"

Response: We appreciate this suggestion from the reviewer. We have restated this sentence to clarify that all survey questions were delivered orally by a member of the research team.

Comment: L163: Although there are no clear cutoff points for DrInC tool, what were the cutoff points used by previous studies that could be used as reference? OR what was the reference used to determine the cutoff points used in the current study?

Response: We appreciate this question from the reviewer and acknowledge this gap in our manuscript. We have updated our section ‘Instruments and Variable Measurements’ to further explain this gap. This change now reads, ‘…while the DrInC tool has been validated in the study setting (DOI: 10.3389/fpubh.2018.00330.), no particular clinically relevant cut-off score for this tool exists, in part because this is among the first studies to apply this tool among Kiswahili-speaking patients, but also because ideally individuals would not experience any alcohol-related consequences. To help account for this lack of cut-off score, DrInC scores were correlated with alcohol use as is detailed further below.’

Comment: L202: Was there any software tool used to analyze the qualitative data?

Response: Thank you for bringing this gap in our methods section to our attention. We have added to our manuscript that ‘All qualitative analysis was conducted in NVivo software version 12 and approved by the entire research team prior to the reporting of these results’ to the end of our qualitative data analysis procedure subsection.

Comment: The discussion part is good, however the perceived harm which is one of the qualitative results was more discussed as compared to perceived benefits. The comparison from different study could provide more information which can be used for future interventions of alcohol use disorders.

Response: We appreciate this comment from the reviewer and acknowledge our discrepancy between discussed harms versus benefits. We have expanded upon our discussion of the perceived benefits of alcohol use, particularly in explaining the effects of increased social unity and financial growth for local sellers, throughout the discussion section. However, in general, IDI participants mentioned substantially more alcohol-related harms than benefits and thus were given greater space in our discussion section overall.

Comment: Also, I would suggest adding different studies which did not concur with the results obtained from the current study so as to bring more challenges to the future researchers.

Response: We have added greater discussion and reference to other studies that challenge our conclusions. For example, one addition reads, ‘While this statement [from the WHO] and our qualitative data reports no physical benefits with alcohol use, past research has indicated that moderate alcohol intake can have positive circulatory and cardiovascular effects as well as being protective against the incidence of type 2 diabetes.’

Academic Editor:

Comment: List the authors before the authors’ affiliation.

Response: Thank you for the feedback. The authors are listed above their respective affiliations on page 1.

Comment: Authors’ contribution has to come at the end.

Response: We appreciate this revision. The authors’ contributions have been moved to page 37 after the manuscript’s conclusion section.

Comment: No need of repeating the title again in page 3

Response: We appreciate this revision. The title has been deleted from page 3.

Comment: Have a heading abstract, then its sub-heading.

Response: We acknowledge this request and have updated the headings and sub-headings for the abstract.

Comment: Conclude the background of abstract with the general purpose of the study.

Response: We appreciate this suggestion from the editor. We have updated the last sentence of our background within our abstract to read, ‘through the voices of Kilimanjaro Christian Medical Center (KCMC) patients, this study aimed to investigate community perceptions surrounding alcohol and the impact of its use in this region.’

Comment: In the method section have a sentence on the data collection method and tool, as well it is not clear on the data analysis method used in the study.

Response: We thank the editor for this comment and have edited the methods section of our abstract accordingly. Our last sentences now read, ‘The impact and perceptions of alcohol use were measured through Drinkers’ Inventory of Consequences (DrInC) sores and analyzed in RStudio using descriptive proportions. IDI responses were explored through a grounded theory approach using both inductive and deductive coding methodologies in NVivo.’

Comment: Conclusion section is not talking on the key findings obtained from the quantitative part of the study, as well in the conclusion what do you mean be ED men and ED women? These were men and women attending emergency department they cannot be called ED men and women

Response: We appreciate this comment from the editor. We have edited the discussion section to include a more thorough discussion of quantitative results and how these in turn interact with our qualitative findings. Additionally, we have also removed “ED men” and “ED women” phrasing from these sections and replaced it with “men and women attending the ED”.

Comment: In line 79, the sentence has language issue, line 112 change the sentence to past tense, this is a report.

Response: We appreciate these comments from the editor and both sentences have been updated accordingly. The first now reads, ‘While hazardous alcohol consumption is a global issue, its use is increasing especially rapidly in low- and middle-income countries.’ The latter sentence has been upda

---

## [Decision Letter · Decision Letter 1]

21 Aug 2024

PONE-D-23-33073R1Understanding the Impacts and Perceptions of Alcohol Use in Northern Tanzania: A Mixed-Methods AnalysisPLOS ONE

Dear Dr. Staton,

Thank you for submitting your manuscript to PLOS ONE. After careful consideration, we feel that it has merit but does not fully meet PLOS ONE’s publication criteria as it currently stands. Therefore, we invite you to submit a revised version of the manuscript that addresses the points raised during the review process.

**Your manuscript has been re-evaluated by two reviewers and their comments are available below. Reviewer 1 has raised further concerns around the methodological reporting that need to be addressed before publication. Please review their comments and make the appropriate revisions to the manuscript.**

We look forward to receiving your revised manuscript.

Kind regards,

Emma Campbell, Ph.D

Staff Editor

PLOS ONE

Reviewers' comments:

Reviewer's Responses to Questions

**Comments to the Author**

1. If the authors have adequately addressed your comments raised in a previous round of review and you feel that this manuscript is now acceptable for publication, you may indicate that here to bypass the “Comments to the Author” section, enter your conflict of interest statement in the “Confidential to Editor” section, and submit your "Accept" recommendation.

Reviewer #1: All comments have been addressed

Reviewer #2: All comments have been addressed

2. Is the manuscript technically sound, and do the data support the conclusions?

Reviewer #1: Partly

Reviewer #2: Yes

3. Has the statistical analysis been performed appropriately and rigorously? 

Reviewer #1: No

Reviewer #2: Yes

4. Have the authors made all data underlying the findings in their manuscript fully available?

Reviewer #1: Yes

Reviewer #2: No

5. Is the manuscript presented in an intelligible fashion and written in standard English?

Reviewer #1: No

Reviewer #2: Yes

6. Review Comments to the Author

**Reviewer #1:**  S/N SECTION DESCRIPTION

1 Introduction • Knowledge gap is not adequately described. The author did not describe what is a real problem under the study

• The author does not indicate any justification of undertaking this study

2 Methods QUANTITATIVE

Study design

• Study design is not clearly described as the author does not illustrate if the study is hospital based or community based because in many areas within the manuscript the authors indicate both patients and community members eg in L# 647. Therefore, the authors should be consistence so as to make it easy understood by readers

Description regarding study sample/population not clear

• In a quantitative part the author stated that “the study team planned to enroll more women the men but the author does not give any justification on that?

Instruments and Variable Measurements

• Description regarding instruments used in the data collection is not clear

• Also the author stated that “DrInC tools has been validated” the author does not indicate the type of validity employed in validation process and unfortunately the DrInC items in description differ with the item in the referred table

• Additionally the author does not tell anything about the reliability of the tool used

• The author does not describe clearly a cutoff points for DrInC tool, what were the cutoff points used by previous studies that could be used as reference? OR what was the reference used by the authors to determine the cutoff points used in the current study?

• Also the author describe AUDIT as tool in the analysis part but does not included in the instrument used description

QUALITATIVE

Sample Size Estimation

• Description regarding sample size estimation in a qualitative part is not clear and sufficient as:

o The author stated that “the study team aimed to enroll 20 ID participants or until data saturation has been met” which one does the author rely on? How sure the author anticipated that the saturation will be reached within 20 participants?

o Which criteria the author used to decide to select 20 participants from 655?

o Also, which criteria does the author use to select 10 men out 0f 114 and 10 women, five from each group? so as to avoid bias?

• Sampling technique

o Does the author take 19 patients from all 676 enrolled patients or 19 from 655 patients who completed survey in quantitative part?

o Which sampling technique the author utilized to select 20 participants out of 655 participants?

• Data Collection and Instruments

o The author does not indicate the sample population used in the pilot of the interview guide

o The author does not illustrate if the participants was informed about the reimbursement provided to them before been interviewed?

• Data Analysis Procedure

o The author does not give the sufficient rationale of using grounded theory in the data analysis

o Why the author does not consider to use thematic analysis as it was before rather than using a grounded theory?

3 Results QUANTITATIVE DATA

• The author just interpret only the demographic characteristics of the participants

• L# 288-291 the author does not indicate any statistical data that support the given interpretation

• Also the alcohol-related consequences or impact as it was the aim of the study should be presented or showed in this quantitative part before taken it into qualitative part but the author does not provide any results regarding those alcohol impact

• The author does not provide the descriptive analysis regarding AUDIT and DrInC score before taking it to find its asssciation

• Additionally in the previous reviewer comments the author responded that “descriptive statistics were used in this analysis with the intention of understanding these populations holistically” but there is no any descriptive analysis performed rather the author performed linear regression to establish the association between AUDIT and DrInC score

• L#295-297 the information described is not been found in the table 2 which the author referring to

• Why does the author include the P-values in the demographic information table?

• The author does not indicates those variables analyzed so as to get these P-value? and which statistical test did he/she use?

• The author should consider to separate the P-Value data from the demographic information table and to have its own table that will shows those variable analyzed

• L# 288-289 the author stated that “Men attending ED found to spend the most of their money on alcohol per week” but unfortunately the author does not shows the data that justify this interpreted results since in demographic information table it indicates the personal income per month and there is no any information indicates how much money spent on alcohol by all three groups of participants so as to reach to that conclusion the author ended with

• L# 205-206 the stated that “Overall and category specific DrInC scores for the three patient populations were assessed through descriptive frequencies and proportions” but on the results section there is no any descriptive frequency and proportion illustrated there. Therefore the authors should revise a section of DATA ANALYSIS Procedure in quantitative so as to tally with the results provided

• The author does not provide any interpretation regarding the information provided in figure 1, figure 2 and in table 3

QUALITATIVE

• The authors does not describe which criteria they use to select 19 participants out of 655 participants for qualitative

• The author did not describe the demographic characteristics of the participants participating in qualitative as not all participants from the quantitative were included on the qualitative. Therefore the author should provide the table showed the characteristic of the participants involved in qualitative part

4 Discussion • The discussion part is good, however the perceived harm which is one of the qualitative results was more discussed than perceived benefits. The author should make comparison from different study conducted in the same area of the study which can be used for future interventions

• L# 531-533 describe that the current study use the existing data. Does the author use the existing data or they conduct survey? The author should revise this part to make it clear

• The author should discuss the findings revealed in the study in relation to other research conducted on the same area

• The author does not discuss the findings for quantitative data, they discuss only the qualitative findings

• Also the author includes the recommendation within the discussion section. It will be more wise, presentable and easily pinpointed by the respective authorities for action if the author separate the recommendation on the separate from discussion with its sub- title before limitation of the study section and it should tally with the results obtained from the study

5 Conclusion • The author should conclude in relation with the revealed results

6 General comments • The manuscript is in Standard English; however, there are numerous typographical and grammatical errors

• The author should justify the works so as to make the work clean with crisp edges

• The author need to make a results for a quantitative clear and understood

• The result from quantitative and qualitative should tally with the title of the study as the current findings based more on the perception of alcohol consumption and live behind the impact of alcohol impacts which was supposed to be revealed and elaborated on the quantitative part so as to provide justification for proceeding with qualitative part of the study

**Reviewer #2: ** The authors have done great work on the previous comments and therefore have addressed all the concerns presented.

7. PLOS authors have the option to publish the peer review history of their article (what does this mean? ). If published, this will include your full peer review and any attached files.

**Do you want your identity to be public for this peer review?** For information about this choice, including consent withdrawal, please see our Privacy Policy .

Reviewer #1: No

Reviewer #2: No

---

## [Author Response · Author response to Decision Letter 2]

14 Dec 2024

December 8th, 2024

To whom it may concern,

We very much appreciate the helpful suggestions and methodological questions raised by Reviewer 1 and thank them for the opportunity to further improve our manuscript. We have incorporated the requested changes in our revised manuscript, which has been attached to this resubmission packet. Our specific modifications are outlined below, and can also be found as a word document in our attached files. We very much appreciate your time and consideration of our work.

INTRODUCTION

Comment: Knowledge gap is not adequately described. The author did not describe what is a real problem under the study

Response: We thank the reviewer for this comment. We have edited the last paragraph of the introduction to better define and outline our knowledge gap for our readers. This change reads: “However, an exploration of the specific societal views of alcohol use within Moshi or the perceived harms and benefits that underlie continued consumption has yet to be conducted. Understanding the full scope of alcohol’s impact and role within the community is essential in combatting excess intake as it allows local leaders and researchers to address the root cause of dependence.” We hope this clarifies the concerns regarding the knowledge gap we aim to fill in this manuscript.

Comment: The author does not indicate any justification of undertaking this study.

Response: We thank the reviewer for this suggestion and the opportunity to improve our manuscript in addressing it. We have strengthened the justification portion of our introduction, which can be found in the last paragraph of the introduction. Our justification now reads, “Understanding the full scope of alcohol’s impact within the community is essential in combatting excess intake as it allows local leaders and researchers to address the root cause of dependence. This knowledge can, in turn, better inform future alcohol-reduction policies, interventions, and research initiatives, allowing them to be more targeted, socioculturally appropriate, and effective in decreasing alcohol use and related harm in Moshi [36]. This process of development can also serve as a model for population-specific harm reduction interventions in other settings.”

METHODS

Quantitative

Comment: Study design is not clearly described as the author does not illustrate if the study is hospital based or community based because in many areas within the manuscript the authors indicate both patients and community members eg in L# 647. Therefore, the authors should be consistent so as to make it easy understood by readers.

Response: We appreciate this feedback from the reviewer. We have modified our “Study Overview” section of the methods to state that this was a hospital-based study. Additionally, we have updated our terminology throughout the introduction, methods, and quantitative results section to describe participants as patients rather than community members. However, the use of the word ‘community’ has been retained within the qualitative portion of our manuscript as patients directly describe the impact of alcohol on their community; for example, IDI #19 states that his “community strongly hates women who drinks alcohol” or IDI #2 says “people who drink Gongo and Dadii are really stigmatized in the community.” In these quotes and framing these statements, we use the term community. We hope these edits and discussion clarify any misunderstandings regarding the study setting and population.

Comment: Description regarding study sample/population not clear. In a quantitative part the author stated that “the study team planned to enroll more women the men” but the author does not give any justification on that?

Response: We thank the reviewer for raising this concern and appreciate the opportunity to clarify. We acknowledge that our initial explanation of the study sample size estimation may have been unclear. In the revised manuscript, we provide the following justification:

“Quantitative sample size was calculated based on the difference in average Alcohol Use Disorder Identification Test (AUDIT) scores between our RHC and ED patient populations, as the prevalence of alcohol use disorder (AUD) was a primary point of interest in the broader study [42]. Based on existing literature and input from local research team members, we hypothesized the prevalence of AUD to be 10% among RHC women, 15% among ED women, and 30% among ED men. Thus, at 80% power and 90% confidence, a sample size of 587 participants was calculated to sufficiently estimate the prevalence of risky drinking across the three subgroups. Given that the expected difference in prevalence was narrower between the two female groups than between male and female groups, the study team initially prioritized enrolling more women to ensure sufficient statistical power to detect differences between the two female populations.

Preliminary data analysis revealed that AUD prevalence was much higher than anticipated (40% in ED women and 45% in ED men). Recalculations of sample size based on these updated prevalence estimates revealed 1,200 patients would be needed to determine differences in proportions between these two groups. While the study timeline and funding could not accommodate the newly estimated sample size of 1200 participants, with IRB approval, the study’s target sample size was increased to enroll as many male and female participants as feasible within the study timeline.”

We hope that this revision provides a clearer justification for our enrollment strategy, specifically addressing the reasoning for prioritizing female enrollment based on narrower expected differences.

Comment: Description regarding instruments used in the data collection is not clear

Response: We appreciate this comment from the reviewer. We have added an additional paragraph to our “Instruments and Variable Measurements” section to provide greater clarity on our instruments. This addition reads: “The quantitative surveys administered to participants consisted of three major components: (1) collection of basic demographic data, such as participants’ age and gender; (2) assessment of typical alcohol use practices, including frequency and quantity of consumption as well as typical expenditure on alcohol; and (3) administration of validated instruments, specifically the AUDIT and DrInC tools. The Alcohol Use Disorders Identification Test (AUDIT) is a 10-item screening tool developed by the World Health Organization to identify hazardous, harmful, and dependent patterns of alcohol use. Scores range from 0 to 40, with higher scores indicating more hazardous consumption and scores of 8 or greater signaling clinically significant risk for adverse alcohol-related health effects. The AUDIT has undergone extensive psychometric and clinical validation across diverse populations and has been specifically adapted and validated in Tanzania through its KiSwahili translation, ensuring cultural and linguistic appropriateness for this context.” Please let us know if further description is needed.

Comment: Also the author stated that “DrInC tools has been validated” the author does not indicate the type of validity employed in the validation process and unfortunately the DrInC items in description differ with the item in the referred table. Additionally the author does not tell anything about the reliability of the tool used.

Response: We thank the reviewer for this feedback regarding the DrInC tool and appreciate the opportunity to improve our manuscript in answering these concerns. We have provided greater explanation of the tools reliability and validity which can be found under the ‘Instruments and Variable Measurements’ section of the Methods. This addition reads, “Prior confirmatory factor analysis conducted in the study setting demonstrated the construct validity of the tool, confirming its five-domain structure as reflective of distinct alcohol-related consequences [43]. Reliability was also supported by high internal consistency, with Cronbach’s alpha exceeding 0.8 across all domains [43]. These findings validate the tool’s suitability amongst KiSwahili-speaking participants, ensuring cultural and linguistic relevance in this context [43].”

Comment: The author does not describe clearly a cutoff points for DrInC tool, what were the cutoff points used by previous studies that could be used as reference? OR what was the reference used by the authors to determine the cutoff points used in the current study?

Response: We appreciate this question from the reviewer and have clarified the issue regarding cutoff points for the DrInC tool. This description reads, "Although the DrInC tool quantifies a range of alcohol-related consequences, it does not measure their severity or frequency, and no universally accepted clinical cutoff score exists. Designed to capture the broad spectrum of alcohol-related issues, the tool has not been associated with specific cutoff scores in previous studies [43–45]. Consistent with this approach, we did not apply a cutoff score in our study, and instead, DrInC scores were analyzed in relation to alcohol use patterns as described below.”

Comment: Also the author describe AUDIT as tool in the analysis part but does not included in the instrument used description.

Response: We appreciate this important comment and have provided greater description of the AUDIT survey instrument. The manuscript has been updated accordingly within the “Instruments and Variable Measurement” section, which has been outlined in a previous response.

Qualitative

Comment: Description regarding sample size estimation in a qualitative part is not clear and sufficient.

Response: Thank you for this feedback. We have thoroughly revised this section to better describe and justify our sample size estimation. This addition reads, “Of the 676 individuals initially enrolled in this study, all 655 patients who had fully completed quantitative surveys were eligible for IDI enrollment. The study team aimed to enroll 20 IDI participants — ten male participants from the ED group and ten female participants, five each from the ED and RHC groups — or until data saturation was reached, whichever occurred first. This approach was chosen to provide equal representation from both genders, while also prioritizing thematic saturation as the primary determinant for concluding data collection.

This original estimated sample size of 20 participants was informed by commonly accepted recommendations in qualitative research, which suggest 15-20 interviews as a general guideline for reaching saturation in relatively homogenous groups [50]. Saturation was also explicitly defined as the point at which no new themes emerged after three consecutive interviews for each subgroup (ED men, ED women, and RHC women). To account for the iterative nature of qualitative research, the team regularly monitored emerging themes throughout data collection and concluded after 19 interviews when saturation was reached across all subgroups. This approach helped balance initial pragmatic estimates of sample size with the flexibility needed to capture all relevant themes.”

Comment: The author stated that “the study team aimed to enroll 20 ID participants or until data saturation has been met” which one does the author rely on? How sure the author anticipated that the saturation will be reached within 20 participants?

Response: We appreciate this question from the reviewer and have edited this section accordingly, as can also be seen in our response above. While we initially targeted 20 participants based on qualitative research guidelines, the study did not rely solely on this initial sample size but prioritized thematic saturation as the primary determinant for concluding data collection. Using a pragmatic initial estimate of 20 participants while retaining some degree of flexibility within our sampe size estimate allowed us to ensure that all relevant themes were captured and provided a more robust criterion for ensuring data adequacy. This sentiment can be seen in the revisions detailed above.

Comment: Also, which criteria does the author use to select 10 men out of 114 and 10 women, five from each group? so as to avoid bias?

Response: We appreciate this comment from the reviewer and have better emphasized within our ‘Sample Size Estimation’ section that this was done to ensure equal gender representation in IDI responses. This change is detailed in our reply above.

Comment: Does the author take 19 patients from all 676 enrolled patients or 19 from 655 patients who completed survey in quantitative part?

Response: We thank the reviewer for this important question. We have updated the ‘Sample Size Estimation’ and Sampling Technique’ portions of our qualitative methods to clarify that only participants with fully completed quantitative surveys were eligible for participation in IDI.

Comment: Which criteria the author used to decide to select 20 participants from 655? Which sampling technique the author utilized to select 20 participants out of 655 participants?

Response: We thank the reviewer for bringing these gap in our manuscript to our attention and have revised the “Sampling Technique” section of our methods to better detail our purposive sampling strategy. This revision reads: “A purposive sampling strategy was employed to ensure representation across key demographic and experiential characteristics from the 655 participants who completed the initial quantitative survey. Participants were purposively selected to ensure diversity across key factors, including age, marital status, education level, occupation, tribe, and religion. Additionally, IDI participants were chosen to reflect varying personal experiences with and perspectives on alcohol use, from patients who only reported neutral or positive past experiences with alcohol, used to drink but are now abstinent, have a close friend or relative that drinks heavily, or suffered significant negative consequences from alcohol. Preliminary quantitative findings also informed participant selection to better explore notable patterns identified in the data. For example, early quantitative analysis indicated that divorced or widowed women had above-average alcohol intake, prompting the inclusion of a recently divorced woman with high alcohol consumption to explore this trend in greater depth.

To ensure a representative sample and minimize potential bias, the characteristics of IDI participants were reviewed monthly by the study lead. Any imbalances or gaps in representation were addressed through adjustments to subsequent participant selection. This iterative process ensured that diverse demographic and experiential perspectives were captured while aligning with the study’s overarching focus.” We hope that this revision satisfactorily addresses the concerns raised by the reviewer.

Comment: The author does not indicate the sample population used in the pilot of the interview guide.

Response: We appreciate this comment from the reviewer and have clarified our phrasing on pilot testing to better illustrate the sample population used. This change now reads, “The interview guide was piloted with the study lead and three KiSwahili-speaking research team members, all of whom were local healthcare professionals and researchers familiar with both the target population and healthcare context. Pilot testing was done to ensure its cultural and linguistic validity prior to the commencement of the interviews.” Please let us know if any further revisions are needed.

Comment: The author does not illustrate if the participants was informed about the reimbursement provided to them before been interviewed?

Response: We appreciate this feedback on our manuscript and the opportunity to improve our work in answering it. We have revised this section to clarify that participants were informed prior to participation in the study that they would be reimbursed for travel to and from KCMC. This clarification reads: “Participants were informed prior to participation that they would be given a small stipend (5000 TSH, equivalent to approximately $2 USD) to reimburse their travel expenses.” We hope this edit resolves any concerns regarding interviewee reimbursemen

---

## [Decision Letter · Decision Letter 2]

16 Jun 2025

PONE-D-23-33073R2Navigating Alcohol’s Impact: A Mixed-Methods Analysis of Community Perceptions and Consequences in Northern TanzaniaPLOS ONE

Dear Dr. Staton,

Thank you for submitting your manuscript to PLOS ONE. After careful consideration, we feel that it has merit but does not fully meet PLOS ONE’s publication criteria as it currently stands. Therefore, we invite you to submit a revised version of the manuscript that addresses the points raised during the review process.

We look forward to receiving your revised manuscript.

Kind regards,

Adetayo Olorunlana, Ph.D.

Academic Editor

PLOS ONE

Journal Requirements:

Reviewers' comments:

Reviewer's Responses to Questions

**Comments to the Author**

1. If the authors have adequately addressed your comments raised in a previous round of review and you feel that this manuscript is now acceptable for publication, you may indicate that here to bypass the “Comments to the Author” section, enter your conflict of interest statement in the “Confidential to Editor” section, and submit your "Accept" recommendation.

Reviewer #1: (No Response)

2. Is the manuscript technically sound, and do the data support the conclusions?

Reviewer #1: Yes

3. Has the statistical analysis been performed appropriately and rigorously? 

Reviewer #1: Yes

4. Have the authors made all data underlying the findings in their manuscript fully available?

Reviewer #1: Yes

5. Is the manuscript presented in an intelligible fashion and written in standard English?

Reviewer #1: Yes

6. Review Comments to the Author

Reviewer #1: The author addressed all comment raised in the previous review but there are some of the comment were addressed inadequately so the author could recheck on it again. After addressing the remaining comment the manuscript would sound better.

7. PLOS authors have the option to publish the peer review history of their article (what does this mean? ). If published, this will include your full peer review and any attached files.

**Do you want your identity to be public for this peer review?** For information about this choice, including consent withdrawal, please see our Privacy Policy .

Reviewer #1: No

---

## [Author Response · Author response to Decision Letter 3]

1 Aug 2025

We appreciate the helpful suggestions and questions raised by Reviewer 1 and thank them for the opportunity to further improve our manuscript. We have incorporated the requested changes. Our specific modifications are outlined below, and our revised manuscript has been attached to this resubmission packet.

INTRODUCTION

Comment: LN# 72-75 The author should rephrase the sentence to make it clear

Response: We thank the reviewer for this suggestion. We have revised Lines 71-74 to more clearly state how alcohol is embedded in social and cultural norms in Tanzania. The revised version now states: “Although alcohol use is associated with significant harms, it remains deeply embedded in cultural and social practices around the world [22]. In Tanzania, for example, alcohol is often normalized from a young age and used as a form of social currency—practices that contribute to higher levels of consumption [23].”

Comment: LN# 91-94. The author addresses this comment insufficiently since the authors report their previous work it could be better if they could indicate the prevalence of alcohol consumption and prevalence of alcohol-related injuries in the area.

Response: We appreciate this comment from the reviewer. We have revised Lines 92-97 to include an additional sentence that reports the prevalence of alcohol-related injuries at KCMC. The revised sentence states, “Among injury patients presenting to the Kilimanjaro Christian Medical Centre Emergency Department, nearly 30% had been consuming alcohol when the injury occurred.” Additionally, we have revised Lines 89-90 to specify that the prevalence of alcohol use is 2.5 times higher than in nearby regions and one of the highest reported rates of alcohol intake in Tanzania. We hope these revisions sufficiently address the reviewer’s concerns.

METHODS:

Comment: Description regarding instruments used in the data collection is not clear

Response: We appreciate this comment from the reviewer. We have revised the quantitative instruments and variable measurements section to more clearly describe the data collection tools used in our study. Specifically, we have clarified the format, items, and scoring of the Drinker Inventory of Consequences (DrInC) assessment in Lines 222-228 stating, “The Drinker Inventory of Consequences (DrInC) was used to quantitatively assess the consequences of alcohol use among participants. This 50-item, yes/no questionnaire has been previously cross-culturally adapted and clinically validated at Kilimanjaro Christian Medical Centre (KCMC) [51]. The tool evaluates alcohol-related consequences across five domains: Physical, Social Responsibility, Interpersonal, Intrapersonal, and Impulse Control (Table 1) [52,53]. Of the 50 items, 45 assess negative consequences and are positively scored, while 5 serve as control items, reflecting perceived benefits of alcohol use (e.g., enjoying the taste or drinking without problems), and are reverse scored. Responses were collected for both the past three months and across the participant’s lifetime.”

Comment: The author stated that" the survey tool used consist 50 items from

5 domains and you refer to table 1 but in the actual table there are 45 items. where are those 5 items? or which item are you referring to?. Also, the author should consider rechecking LN# 211-211 which table are you referring? Since the actual Table 1 there are 45 items and not 50 as the author started.

Response: We appreciate this important question from the reviewer. We have revised Lines 222-286 to more clearly articulate the structure, subcategories, and scoring of the DrInC tool. Specifically, the five subcategories of the DrInC tool contain 45 items that measure adverse consequences related to alcohol use, which are depicted in Table 1. An additional five control items that reflect perceived alcohol-related benefits are included to comprise 50 total items. All 50 items were assessed in the study and scored appropriately (45 items positively scored, 5 items reverse scored).

Comment: The author does not indicate the sample population used in the pilot of the interview guide

Response: We thank the reviewer for this comment. We have revised Lines 324-328 to more clearly describe our sample population for the qualitative interview pilot. Notably, we have added the clinical and research backgrounds of the KCMC research assistants who the guide was piloted with and their role in ensuring the cultural and linguistic validity of the interview guide.

Comment: What was the sample used in the pilot? is the same sample included in the actual study? it would be better if you indicated the sample used in the pilot. are they the same patients used in the study? or? are they from the same clinical departments?

Response: We thank the reviewer for this thoughtful comment regarding the pilot sample. We have revised the manuscript to explicitly describe the individuals involved in piloting the interview guide. This addition reads: “The interview guide was piloted with the study lead and three Kiswahili-speaking research team members from the research department at Kilimanjaro Christian Medical Centre (KCMC). These team members, who were not part of the final study sample, had professional backgrounds in nursing, medicine, and qualitative research, and were familiar with the target population and healthcare setting. Their feedback informed refinements to ensure the cultural and linguistic validity of the guide prior to initiating interviews with participants. Pilot interviews were conducted solely for refinement purposes and were not included in the study dataset.”

As noted, the pilot was conducted with three Kiswahili-speaking research team members at KCMC with professional experience in nursing, medicine, and qualitative research. Although these individuals were not part of the study sample, their familiarity with the local healthcare context and the target population allowed them to provide meaningful feedback on both linguistic clarity and cultural relevance. The pilot interviews were used exclusively to refine the guide and were not included in the study data or analysis. While we acknowledge that piloting directly with members of the target population can offer additional advantages, this approach was not conducted with this study. Nonetheless, we are confident that our piloting process appropriately ensured the guide’s validity and suitability for use in this context.

GENERAL COMMENTS:

Comment: The authors address all comments adequately except for the remaining few which were inadequately addressed and it would be wise if the author could address them to make the manuscripts better.

Response: We thank the reviewer for this comment and the opportunity to enhance our work. We hope our revisions have adequately addressed the reviewer’s comments.

Comment: The author should justify the work to make the work clean with crisp edges. This is about the alignment of the work and not justification of work as the author tries to address this comment.

Response: We appreciate this comment from the reviewer. We have made two changes to the manuscript to address the reviewer’s suggestions. First, we have revised Lines 106-107 of the introduction section to more clearly justify the work and the knowledge gap. The revised text states, “The justification of this work lies in its potential ability to inform future alcohol-reduction policies, interventions, and research initiatives, allowing them to be more targeted, socioculturally appropriate, and effective in decreasing alcohol use and related harm in Moshi.” This statement helps to emphasize the broader alignment of this work towards alcohol-reduction strategies and improving health outcomes.

Second, we have included an additional sentence in Lines 726-728 of the discussion section to articulate how our research expands and aligns with existing alcohol research in the region. The revised text states, “Building on existing alcohol research in the region, insights from this analysis can help guide future culturally relevant research and alcohol reduction efforts in Moshi, and inform similar interventions in other communities.” We hope these revisions adequately address the reviewer’s concerns.

---

## [Decision Letter · Decision Letter 3]

17 Aug 2025

Navigating Alcohol’s Impact: A Mixed-Methods Analysis of Community Perceptions and Consequences in Northern Tanzania

PONE-D-23-33073R3

Dear Dr. Staton,

We’re pleased to inform you that your manuscript has been judged scientifically suitable for publication and will be formally accepted for publication once it meets all outstanding technical requirements.

Kind regards,

Adetayo Olorunlana, Ph.D.

Academic Editor

PLOS ONE

Additional Editor Comments (optional):

Reviewers' comments:

Reviewer's Responses to Questions

**Comments to the Author**

1. If the authors have adequately addressed your comments raised in a previous round of review and you feel that this manuscript is now acceptable for publication, you may indicate that here to bypass the “Comments to the Author” section, enter your conflict of interest statement in the “Confidential to Editor” section, and submit your "Accept" recommendation.

Reviewer #1: All comments have been addressed

2. Is the manuscript technically sound, and do the data support the conclusions?

Reviewer #1: Yes

3. Has the statistical analysis been performed appropriately and rigorously? 

Reviewer #1: Yes

4. Have the authors made all data underlying the findings in their manuscript fully available?

Reviewer #1: Yes

5. Is the manuscript presented in an intelligible fashion and written in standard English?

Reviewer #1: Yes

6. Review Comments to the Author

Reviewer #1: The author address all the comments therefore editorial team can proceed with the publication process

7. PLOS authors have the option to publish the peer review history of their article (what does this mean? ). If published, this will include your full peer review and any attached files.

**Do you want your identity to be public for this peer review?** For information about this choice, including consent withdrawal, please see our Privacy Policy .

Reviewer #1: No

---

## [Editor Report · Acceptance letter]

PONE-D-23-33073R3

PLOS ONE

Dear Dr. Staton,

I'm pleased to inform you that your manuscript has been deemed suitable for publication in PLOS ONE. Congratulations! Your manuscript is now being handed over to our production team.

Kind regards,

on behalf of

Associate Professor Adetayo Olorunlana

Academic Editor

PLOS ONE